# Study on Bionic Design and Tissue Manipulation of Breast Interventional Robot

**DOI:** 10.3390/s24196408

**Published:** 2024-10-03

**Authors:** Weixi Zhang, Jiaxing Yu, Xiaoyang Yu, Yongde Zhang, Zhihui Men

**Affiliations:** 1Key Laboratory of Advanced Processing Technology and Intelligent Manufacturing (Heilongjiang Province), Harbin University of Science and Technology, Harbin 150080, China; jidianzhangweixi@163.com (W.Z.); jidianyujiaxing@163.com (J.Y.);; 2Postdoctoral Research Center of Instrument Science and Technology, Harbin University of Science and Technology, Harbin 150080, China

**Keywords:** breast interventional robot, bionic design, ADMM-based tissue dynamics, breast tissue manipulation, manipulate target displacement

## Abstract

Minimally invasive interventional surgery is commonly used for diagnosing and treating breast cancer, but the high fluidity and deformability of breast tissue reduce intervention accuracy. This study proposes a bionic breast interventional robot that mimics the scorpion’s predation process, actively manipulating tissue deformation to control target displacement and enhance accuracy. The robot’s structure is designed using a modular method, and its kinematics and workspace are analyzed and solved. To address the nonlinear breast tissue deformation problem, a hierarchical tissue method is proposed to simplify the three-dimensional problem into a two-dimensional one. A two-dimensional tissue deformation solver is established based on the minimum energy method for quick resolution. The problem is treated as quasi-static, deriving the displacement relationship between external manipulation points and internal tissue targets. The method of active manipulation of tissue deformation is simulated using MATLAB (2019-b) software to verify the feasibility of the method. Results show maximum errors of 1.7 mm for prostheses and 2.5 mm for in vitro tissues in the *X* and *Y* directions. This method improves intervention accuracy in breast surgery and offers a new solution for breast cancer diagnosis and treatment.

## 1. Introduction

Global cancer statistics released by the International Agency for Research on Cancer (IARC) of the World Health Organization show that breast cancer accounts for as many as 2.3 million people in the female population, surpassing lung cancer as the number one cancer afflicting women [1,2]. Early acceptance of breast cancer treatment by patients can increase the cure rate and improve the prognosis. Minimally invasive interventions are one of the most common surgical procedures for breast cancer diagnosis and treatment, such as breast biopsy, radioactive particle implantation, and ablation therapy [3]. Minimally invasive interventional diagnosis and treatment uses a needle to remove tissue samples for diagnosis and to intervene at the target point inside the tissue for treatment. If the needle tip deviates during this process, it will lead to misdiagnosis or tissue damage. Therefore, ensuring that the needle tip accurately pierces the target point is the key to interventional surgery. Because of the high mobility and easy deformation of breast tissue, the target point may deviate from the original position, and the blood vessels and nerves inside the breast tissue may also hinder the ideal needle path, complicating the interventional environment [4], as shown in Figure 1.

With the development of medical robotics technology, robots have solved many medical problems. Among them, interventional robots have made many remarkable achievements in the field of interventional surgery with their high-precision interventional technology, low risk of postoperative complications, and reduced pain during surgery. Therefore, it is necessary to develop interventional robots to assist or replace surgeons in performing breast interventional surgery in complex environments. Jiang et al. proposed a novel tendon-driven magnetic resonance-guided robotic system for minimally invasive breast surgery, which had five degrees of freedom (DOFs) [5]. Du et al. proposed a seven-DOF compact breast interventional robot suitable for a Magnetic Resonance Imaging(MRI) environment [6]. Groenhuis et al. proposed an MR safe robotic system for breast biopsy, which could achieve a high-precision intervention in an MR environment [7]. Welleweerd et al. describe an end-effector for robotic 3D ultrasound breast acquisition and US-guided breast biopsy. The end-effector, combined with a seven-DOF robot, manipulated the pose of the needle in order to assist physicians in achieving a high-precision intervention [8]. The robot they designed required the patient to lie prone on the bed and only considered reducing interventional errors caused by tissue deformation due to breathing, while ignoring the problem of easy tissue deformation during surgery. Liu et al. proposed an MRI cable-driven breast interventional robot, which included a device for limiting breast tissue deformation and a three-DOF intervention mechanism [9]. The device for limiting breast tissue deformation was a hemispherical mesh stent. During operation, the diseased breast tissue was placed in the stent, and the healthy tissue was located outside the stent. Finally, the needle was inserted through the three-DOF intervention mechanism. Jácobo-Zavaleta et al. proposed a five-DOF breast interventional robot based on ultrasound guidance [10]. In addition to the intervention mechanism, the robot was designed with a holder device that limited the deformation of breast tissue based on the patient lying prone. Song et al. proposed an integrated navigation system based on a dedicated breast support device (DBSD) to assist physicians in interventional surgery, a gridlike fixation device that limited breast tissue deformation, with the aim of reducing tissue deformation and improving interventional accuracy [11]. They only focused on achieving the target intervention in complex environments by adjusting the needle posture based on fixed breast tissue. However, when intervening at the target site in a complex environment, there is a possibility that no matter how the needle’s posture is adjusted, the safe tissue may be damaged or there may not be a good intervention path. Cao et al. developed a flexible tactile sensor with an embedded hair structure that could decouple normal stress and shear stress and process them through a simple addition and subtraction algorithm with an error within 3% [12]. Planning the needle insertion path and designing the interventional robot that manipulates the flexible needle can solve the problem of target intervention in complex environments [13,14]. However, in clinical breast interventional surgery, the movement path of the interventional needle in the tissue is relatively short, generally about 50 mm. When the insertion length of the interventional needle is 50 mm [15], the maximum deflection of the needle tip is less than 1 mm. Therefore, there are certain limitations in manipulating the deflection of the flexible needle inside the breast tissue. Currently, breast intervention robots are capable of minimizing damage to healthy tissues and performing high-precision interventional surgeries in complex environments, addressing a critical research challenge that urgently needs to be solved.

The method of placing the patient prone on a bed and fixing the breast tissue combined with robotic needle insertion is common, but this method still cause damage to healthy tissue and has low intervention accuracy when intervening at the target site in a complex environment. Although tissue deformation can cause the target to shift, by actively manipulating tissue deformation, the target can be transferred to a simple and safe intervention path, thereby improving intervention accuracy. In clinical surgery, the doctor manipulates and fixes the deformation of breast tissue with one hand and inserts the needle with the other hand. Although clinicians are already familiar with methods of actively manipulating breast tissue deformation to shift targets and improve the accuracy of interventional surgery, doctors’ active manipulation of tissue deformation and control of the insertion of interventional needles have problems such as a long target shifting time, low intervention accuracy, and reliance on doctor experience. Bionic technology is increasingly used in the field of minimally invasive medical care. We noted that the scorpion’s predation process is similar to clinical breast interventional surgery. Inspired by the scorpion’s predation process, this paper designs a bionic breast interventional robot that is consistent with the clinical surgical process. The robot can manipulate breast tissue deformation on behalf of doctors, shift the target to the ideal position, and insert the needle, overcoming problems such as the difficulty of target intervention in complex environments, low intervention accuracy, and reliance on doctor experience.

The goal of this paper was to develop a breast interventional robot that mimics the scorpion’s predation process. The robot has both a puncture module and a tissue manipulation module. After establishing the tissue deformation solver, the tissue deformation problem is regarded as a quasi-static problem, the displacement relationship between the target and the manipulation point is further derived, and PID control is used. MATLAB is used for simulation, and experiments on breast prostheses and in vitro tissues are conducted to verify its feasibility. The core of this method is to use the suction cup of the tissue manipulation module to actively manipulate the deformation of breast tissue, so that the target is eventually displaced to a safe and simple puncture path.

The remainder of this paper is organized as follows. Section 2 describes the bionic design and analysis of the breast interventional robot. Section 3 elaborates on the dynamic model for resolving breast tissue deformation, the neo-Hookean model incorporating fiber characteristics, and the linearization of the model under quasi-static conditions. This resolves the displacement relationship between the manipulation and target points. An experimental platform is built in Section 4 to experimentally validate tissue manipulation. Section 5 concludes the paper with a summary and outlook.

## 2. Structural Design and Analysis of Breast Interventional Robot

### 2.1. Demand Analysis of Breast Interventional Robots

In actual clinical surgery, surgeons usually use one hand to fix and control the deformation of breast tissue, and the other hand to adjust the posture of the needle to complete the needle insertion operation. Therefore, this study modularly designed the breast intervention robot based on the actual operation mode of the doctor during surgery. The robot consisted of a puncture module and a tissue control module, and the integration of the medical image system was not considered. In order to achieve the robot’s precise operation of breast tissue and accurate positioning of the puncture target, these two modules needed to meet specific degrees of freedom requirements, as shown in Figure 2.

(1)In order to ensure that the needle tip can cover the entire breast, the puncture module needs to adjust the position of the needle tip in the three-dimensional rectangular coordinate system, that is, a linear movement in the *x*, *y*, and *z* directions, which requires 3 DOFs. In addition, in order to simulate the doctor’s operation in clinical surgery, the rotation and linear insertion functions of the needle must also be considered, which require 1 DOF each, respectively. Therefore, the puncture module requires at least 5 DOFs to meet basic puncture requirements.(2)In order to simulate the doctor’s fixation and manipulation of breast tissue, the tissue control module needs to have the function of fixing the breast and manipulating its position. When operating a single breast, the module needs 1 DOF to fix the tissue and another DOF to push and pull the tissue. Therefore, the tissue control module needs at least 2 DOFs to meet basic control requirements.(3)Interventional needles for breast interventional surgery are divided into fine and coarse needles, which are mainly used for the initial determination of tumor benignity, and the coarse needles are hollow inside. In breast intervention surgery, 18G interventional needles are generally used. Those with an outer diameter of 1.2 mm belong to the range of coarse needles, and 18G interventional needles can be used for the final determination of tumor benignity and targeted therapy.

### 2.2. Bionic Concept and Design of Breast Interventional Robot

Bionic design is increasingly being used in the medical field. Scorpions, unlike other animals, possess two different weapon systems: the anterior chelae and the pedipalp with the sting [16]. The hunting process is as follows: When a scorpion catches its prey, it first grabs the prey with the anterior chelae. Then, it uses the anterior chelae to adjust the position of the prey until it reaches an angle where the pedipalp with the sting can easily fire its poisonous sting. Finally, it adjusts the posture of the pedipalp with the sting to accurately stab the prey.

Inspired by the principle of scorpion predation, a bionic breast interventional robot that could actively manipulate tissues was designed. The robot’s tissue manipulation block imitated the function of the anterior chelae of the scorpion to manipulate and fix breast tissue, and the puncture module imitated the function of the pedipalp with the sting of the scorpion to flexibly adjust the needle posture for intervention. The 5-DOF puncture module described in Section 2.1 lacks the ability to adjust the needle deflection path and cannot flexibly change the needle insertion path. The arc telecentric motion Remote Center of Motion (RCM) mechanism is commonly used in medical surgical robots because it can rotate around the distal point, making it easier to adjust the position and posture of the needle [17]. The advantage of the arc RCM is that it can achieve needle deflection while keeping the needle tip position unchanged, thereby adjusting the needle insertion path. Compared with the traditional deflection mechanism, the arc RCM allows the needle insertion path to be flexibly adjusted by deflecting the needle after the needle insertion point is determined, without changing the position of the needle tip. Therefore, based on the basic requirements of the puncture module in Section 2.1, the arc telecentric motion (RCM) mechanism was introduced to enable the module to better simulate the scorpion’s chelicerae operation mode and realize the path deflection of the needle. Since the arc RCM added 1 DOF, the degree of freedom of the puncture module increased from the original 5 DOFs to 6 DOFs. In addition, although the tissue control module could fix and manipulate breast tissue, it could not adjust the position of the manipulation point. To this end, it was necessary to add 1 DOF to rotate around the breast, so that when operating a single breast, the degree of freedom of the tissue control module increased from the original 2 DOFs to 3 DOFs, as shown in Figure 3. This paper studied the use of 18G rigid needles in breast interventional surgery. In order to reduce the intervention error caused by breathing during surgery, patients usually lie prostrate on the bed to minimize the target position error caused by breathing. The maximum static chest width of women aged 18 to 70 is about 335 mm, and the height of the breast is about 100 mm [18]. Therefore, the working space of the puncture module must reach or exceed the size of 150 × 330 × 100 mm to meet the surgical requirements, as shown in Figure 4.

The bionic breast interventional robot that mimics the scorpion’s predation process was divided into a puncture module and a tissue manipulation module, as shown in Figure 5. The robotic puncture module mainly included a right-angle coordinate mechanism, arc RCM telecentric positioning mechanism, and needle insertion mechanism. The rectangular coordinate mechanism was composed of three ball screw modules, which could adjust the *x*, *y*, and *z* directions of the rectangular coordinate system. The ball screw module in the *z* direction used a synchronous belt to place the 28-step motor in a more stable position. The arc-shaped RCM telecentric positioning mechanism was connected to the slider of the *z*-axis lead screw module. The motion of the arc RCM mechanism was transmitted through a pair of large and small gears. The large gear was halved and was connected to the *z*-axis lead screw module, which served as the track for the small gear to move. The small gear meshed with the half large gear and was driven by the motor on the RCM motor support seat. The small gear drove the RCM motor support seat to move forward along the track of the large gear. The RCM motor support seat was also connected to the needle insertion mechanism, driving the needle insertion mechanism forward. The 18G rigid needle was connected to the rotating electrical machines and fixed in the nut battle of the needle insertion mechanism. The needle insertion mechanism realized the insertion and rotation of the needle. It is worth noting that the center of the arc-shaped large gear track (the distal point) needed to coincide with the position of the needle tip point to achieve the immobility of the needle tip and the circumferential adjustment of the needle body around the distal point. The tissue manipulation module consisted of a preloading platform, turntable module, linear ball-screw slide, and suction cup module. The suction cup was fixed to the slide on the linear ball screw. Two linear ball-screw slides were placed vertically to fix and pull down the breast tissue, whereas the other two slides were mounted on the turntable and could be adjusted 180° to flexibly position the manipulation boundary points of the breast tissue. The rotation of the turntable module was driven by a pair of large and small gears. The large gear was connected to the turntable and could rotate with the large gear. This design not only improved the flexibility of tissue manipulation but also reduced the space required for the tissue manipulation module and avoided interference of the robot during work. Vertical and horizontal manipulations were achieved by absorbing the breast tissue with vertical and horizontal suction cups.

The physical prototype of the robot was made of poly lactic acid (PLA), aluminum, copper, and iron. Since the manipulation of a single breast and two breast tissues has the same meaning, this paper only studied a single breast tissue to verify the feasibility of the theory. The robot’s workflow steps were as follows: First, the patient lies prone on the bed. Second, the medical imaging system determines the position of the target inside the tissue. Then, the data are fed back to the robot through a computer, and the robot begins to manipulate the displacement of the target inside the tissue until it reaches the ideal needle insertion path, stops manipulation, and fixes the tissue. Finally, the robot’s puncture module adjusts the needle’s posture to pierce the target, as shown in Figure 6.

### 2.3. Kinematic and Workspace Analysis of Breast Interventional Robots

Since the tissue manipulation module only studied a single breast, and 1 DOF was used to fix the breast, 1 DOF was used to push and pull the breast, and only 1 DOF was used to adjust the position of the manipulation point, this paper only conducted a kinematic analysis on the robot’s puncture module. In this section, the modified DH method was used to study the kinematics of the “puncture module”; the parameters are defined in Table 1 [19].

The puncture module had 6 DOFs, and the rotational freedom of the interventional needle did not affect the adjustment of its end attitude. Therefore, the rotational freedom of the interventional needle was not considered at first. A schematic of the puncture module structure is shown in Figure 7a. The simplified robot had five joints and six links. First, a coordinate system was established for the simplified puncture module model. Because the puncture module had five joints, six coordinate systems had to be established: the coordinate system of the base of the puncture module was {0}, and the coordinate system on the corresponding connecting rod *i* was {*i*} (*i*: 0–6). The joint coordinate system of the puncture module is illustrated in Figure 7b.

Table 2 was obtained based on the established reference coordinate system of the puncture module combined with the improved DH method.

With each parameter in Table 2, the transformation matrix Tii−1 from coordinate system {*i* − 1} to {*i*} can be computed by matrix operations for the position of the end of the puncture module in spatial coordinates:(1)Tii−1=Rotxi−1(αi−1)×Transxi−1(ai−1)×Rotzi(θi−1)×Transzi−1(di)

Simplifying Equation (1) gives
(2)Ti    i−1=cosθi−sinθi0ai−1sinθicosαi−1cosθicosαi−1−sinαi−1−disinαi−1sinθisinαi−1cosθisinαi−1cosαi−1dicosαi−10001

Bringing the parameters of Table 2 into Equation (2) yields
T10=10000100001d10001T21=0100001d2−10000001T32=0100001d3−10000001T43=c4−s40a30010−s4−c4000001T54=0100001d510000001T65=100a50100001d60001

Among them, $c_{i}$ = $\cos\theta_{i}$, $s_{i}$ = $\sin\theta_{i}$.

$a_{5}$ and $d_{6}$ are constants, and T65 is a fixed matrix. To obtain the matrix between the puncture needle coordinate system {6} and the base {0} at the end of the puncture module, the individual Ti    i−1’s are multiplied one at a time.
(3)T60=10T2lT32T43T54T65T=nxoxaxpxnyoyaypynzozazpz0001

The equality of the corresponding elements in the matrix can be obtained:$$\begin{array}{c} n_{x}=0,n_{y}=0,n_{z}=-1,\\ o_{x}=-s_{4},o_{y}=-c_{4},o_{z}=0,\\ a_{x}=-c_{4},a_{y}=s_{4},a_{z}=0,\\ p_{x}=a_{1}+d_{3}-c_{4}*d_{5}-c_{4}*d_{6},\\ p_{y}=d_{2}-a_{3}+s_{4}*d_{5}+s_{4}*d_{6},\\ p_{z}=d_{1}-a_{5}. \end{array}$$

Geometric and analytical methods are typically used for inverse motion analysis of robots. In order to simplify the inverse kinematics solution, the first three kinematic joints were not considered to simplify the process. This solution only analyzed kinematic joints 4, 5, and 6. The mechanism consisted of three key components: the arc RCM structure driving joint (joint 4), the linear needle drive joint (drive needle insertion, joint 5), and the rotation joint around the needle tip (joint 6). It consisted of joint 4, joint 5, and a rotational joint 6 around the tip of the needle, which was solved for $\theta_{4}$ and $d_{5}$. $d_{6}$ was constant. Because the *XYZ* module of the Cartesian coordinates was located when the coordinates of the end of the puncture were confirmed, the matrix A30 of joint 3 with respect to joint 0 after locking the *XYZ* module was constant, which could be changed from the original end-joint coordinates $\left(p_{x},p_{y},p_{z}\right)$ to the end-joint coordinates $\left(p_{x}^{\prime },p_{y}^{\prime },p_{z}^{\prime }\right)$ relative to locking joint 3, set the position of translation probe into joint 6 relative to joint 3 to A63=(px′,py′,pz′,1)T, and also according to A60=A30*A63, could be obtained from the following equation:(4)px′py′pz′1T=A30−1pxpypz1T

The position of the end coordinates relative to joint 3 can be obtained with
(5)px′py′pz′1T=A43A54A650001T

Among them,
(6)A43A54A65==BaxBbxBcxpx′BayBbyBcypy′BazBbzBczpz′0001

In the formula, each item in the matrix is the result of multiplying A43A54A65. We do not expand them one by one here. We can obtain them through calculations in MATLAB:$$\begin{array}{l} p_{x}^{\prime }=a3-d5*s4-d6*s4\\ p_{y}^{\prime }=a5\\ p_{z}^{\prime }=-c4*d5-c4*d6 \end{array}$$

Based on the actual measurement of the mechanism model, we obtained the known parameters and used MATLAB (2019-b) to solve the workspace. The results are shown in Figure 8. The left and right figures show the projection of the breast on the *xoz* plane and the *xoy* plane, respectively. The blue area indicates the spatial position that the needle tip of the puncture module can reach. It can be seen from the figure that the needle tip of the puncture module can completely cover the operating range of the breast on the *xoy* plane, and at the same time, on the *xoz* plane, the workspace is large enough to meet the thickness requirements of the breast. This ensures that the needle tip can accurately cover all areas of the breast and meet the high requirements for the needle tip position during the operation, thereby improving the accuracy of the puncture and the flexibility of the operation.

## 3. Implementation of Breast Tissue Manipulation

### 3.1. Modeling Breast Tissue Dynamics

Rapid modeling of breast tissue deformation is essential to ensure the real-time performance of the operation. Therefore, this paper aimed to develop a solver that could efficiently compute tissue deformation. In the second section, it is emphasized that patients need to lie prone during surgery and the breast tissue is pulled down by the vertical suction cup to limit fluidity. Therefore, this paper proposes a method to combine breast tissue into multiple two-dimensional planes, ignoring the influence of the tissue in the vertical direction, as shown in Figure 9. The meshing in two dimensions is mainly triangular and quadrilateral cells because triangular cells are easy to mesh and adaptable. Therefore, triangular cells were used for meshing. In the next study, we will treat the layered two-dimensional planes as rectangles for research, but in this paper, we only needed to solve the two-dimensional rectangle divided by the triangular unit mesh to obtain the corresponding tissue deformation.

Consider the vertex of each triangle element as a mass matrix M, and each mass has its corresponding DOF. Because the problem studied in this paper focuses on breast tissue deformation, breast tissue also generates internal forces when subjected to a force situation, which is referred to here as an internal force. The external force is pushed by the suction cup, which is considered an external force, and the breast tissue deformation force situation consists of the external force and internal force. According to Newton’s second law, the kinetic equation in the Eulerian framework is as follows:(7)$$M\ddot{x}=F_{ext}+F_{int}$$
where $F_{int}$ denotes the internal force; $F_{ext}$ denotes the external force; and x denotes the displacement vector.

In continuum mechanics, the internal force $F_{int}$ of Equation (7) is obtained as the negative gradient of the strain energy function.
(8)$$F_{int}=-\nabla U(x)$$

U(x) describes the state of the energy change caused by the force inside the breast tissue. Different materials have different strain energy functions; therefore, we need to choose an isomorphic model suitable for the study of deformation of breast tissue, and thus determine the strain energy function, which is determined subsequently.

Now, we must solve Equation (7). We carry this out based on time, so we need to integrate the time, because the implicit Euler integral is stable and unconditional; therefore, we solve Equation (7) using the implicit Euler integral:(9)$$\begin{array}{l} x(t+\Delta t)=x(t)+v(t+\Delta t)\Delta t\\ Mv(t+\Delta t)=Mv(t)+\Delta tf(v(t+\Delta t),x(t+\Delta t),t)\\ =Mv(t)+F_{ext}(t)\Delta t+F_{int}(t+\Delta t)\Delta t \end{array}$$
where Mv(t) is the momentum of the object at time t, and $x(t)\in R^{d}$ is the position of the object moving at time t. Here, $R^{d}$ represents the d-dimensional Euclidean space where the object position vector x(t) is located, and d represents the dimension of the vector, that is, the position description of the system in 2 dimensions.

It is possible to solve the nonlinear system of kinetic Equation (9) by means of iterative algorithms, such as the Newton–Raphson method or the proposed Newton method, and it is also possible to carry out more accurate and complex calculations by means of the Runge–Kutta method; however, these methods are advantageous for dealing with rigid systems and are not suitable for solving them in the breast model. Therefore, Equation (9) was changed to the following expression:(10)$$\begin{array}{l} v(t+\Delta t)=\frac{x(t+\Delta t)-x(t)}{\Delta t}\\ \frac{1}{\Delta t^{2}}M(x(t+\Delta t)-\tilde{x}(t+\Delta t))=F_{int}(t+\Delta t) \end{array}$$
where $\tilde{x}(t+\Delta t)$ is the predicted state at time t+Δt when there is no internal force: $\tilde{x}(t+\Delta t)=x(t)+v(t)\Delta t+M^{-1}F_{ext}(t)\Delta t^{2}$.

Combining it with $F_{int}=-\nabla U(x)$ turns Equation (10) into an optimization problem [20]:(11)$$x(t+\Delta t)=\underset{x}{argmin(}\frac{1}{2\Delta t^{2}}||x-\tilde{x}(t+\Delta t)|{|}^{2}M+U(x))$$
where $||x|{|}_{M}=\sqrt{x^{T}Mx}$, $\tilde{x}(t+\Delta t)$ simplifies to $\tilde{x}$, and x(t+Δt) simplifies to x.

Considering the gradient in Equation (11) and setting it to zero, it was found to be the same as in Equation (9), thus the solution of Equation (11) for each time step is a solution to Equation (9). If traditional solution methods such as gradient descent optimization are used on Equation (11), it consumes a lot of time; instead, it can be solved using Overby et al.’s method [20], who proposed an Alternating Direction Method of Multipliers (ADMM)-based optimization problem. Before adopting the ADMM optimization problem, the strain energy function of the breast tissue must be determined [21].

### 3.2. Modeling of Breast Tissue

Breast tissue has significant nonlinearity and anisotropy (different physical properties in different directions), and its deformation occurs gradually over time after being stressed, which shows that its deformation behavior is closely related to time and exhibits viscoelastic properties. Therefore, a simple mass–spring damping model cannot accurately describe its deformation, and a hyperelastic model must be used to better simulate its stress–strain relationship (stress is the applied force, and deformation is the resulting deformation). There are many types of hyperelastic models, such as neo-Hookean, Mooney–Rivlin, Ogden, Arruda–Boyce, and polynomial [22]. The intrinsic model of hyperelastic materials is defined by the relationship between the total stress and total strain derived from the strain energy function, which can be expressed as follows:(12)$$G=G\left(C\right)$$
where $C=FF^{T}$ is the right Cauchy–Green deformation tensor, and F is the deformation gradient matrix, calculated by
(13)$$F=\partial x^{1}/\partial x^{0}$$
where $x^{1}$ is the post-deformation position of the cell node, and $x^{0}$ is the undeformed position of the cell node.

The stress–strain relationship is derived from Equation (12) as:(14)S=2∂G/∂C
where S is the second Piola–Kirchhoff stress tensor.

In general material modeling, for isotropic materials, we have:(15)$$U=U\left(I_{1},I_{2},I_{3}\right)$$
where $I_{1}=trC$, $I_{2}=\frac{1}{2}[{\left(trC\right)}^{2}-tr(C^{2})]$, $I_{3}=\det C=J^{2}$, J is used to describe the change in volume during material deformation. $I_{3}=1$ for incompressible materials.

As breast tissue includes ligaments and connective tissues, which are viewed as fibrous materials aligned in a specific direction, the inclusion of $I_{4}$ allows for a more realistic simulation of breast tissue deformation. Equation (15) can be rewritten as [23,24]
(16)$$U=U\left(I_{1},I_{2},I_{3},I_{4}\right)$$

Because the neo-Hookean model is the most relevant and commonly used model for breast tissue, we used the neo-Hookean model for further study. Considering that the breast is reinforced by a family of fibers aligned in a specific direction, the strain energy function can be decomposed into an isotropic component and a transverse isotropic component. The total strain energy function is expressed as follows:(17)$$\begin{array}{l} U=U_{1}+U_{2}\\ U_{1}=\frac{\mu }{2}(I_{1}-3)+\frac{k}{2}{\left(J-3\right)}^{2}\\ U_{2}=\frac{\eta }{2}{\left(I_{4}-3\right)}^{2} \end{array}$$
where $J=\frac{1}{2}I_{3}$, $I_{1}=tr(J^{\frac{-2}{3}}C)$, $\mu =\frac{E}{2(1+v)}$, $k=\frac{E}{3(1-2v)}$, η defines the strength of the fiber reinforcement, where E is Young’s modulus, and v is Poisson’s ratio.

### 3.3. ADMM Solves the Kinetic Equations for Breast Tissue Deformation

Equation (11) is expressed using the ADMM as:(18)$$\begin{array}{l} \underset{x,z}{argmin}(\frac{1}{2\Delta t^{2}}||x-\tilde{x}|{|}_{M}^{2}+U(z))\\ s.t.W(Dx-z)=0 \end{array}$$
where $\frac{1}{2\Delta t^{2}}||x-\tilde{x}|{|}_{M}^{2}$ is an item in the multiplicative alternating direction method. The strain energy function was determined in the previous section, where z=Dx, to convert the *x* variable to the gradient deformation matrix space, which is a key part of the solution using the ADMM algorithm, and W is the weighting matrix. By solving Equation (18) using the ADMM algorithm [20], the updated equation for the tissue deformation dynamics’ problem can be obtained as:(19)$$\begin{array}{c} x^{n+1}=\underset{x}{argmin}\left(\frac{1}{2\Delta t^{2}}∥x-\tilde{x}{∥}_{M}^{2}+\frac{1}{2}W{\left(Dx-z^{n}+u^{n}\right)}^{2}\right)\\ ={\left(M+\Delta t^{2}D^{T}W^{T}WD\right)}^{-1}\left(M\tilde{x}+\Delta t^{2}D^{T}W^{T}W\left(z^{n}-u^{n}\right)\right)\\ z^{n+1}=\underset{z}{argmin}\left(U(z)+\frac{1}{2}W{\left(Dx^{n+1}-z+u^{n}\right)}^{2}\right)\\ u^{n+1}=u^{n}+Dx^{n+1}-z^{n+1} \end{array}$$

The matrix in Equation (19) is fixed and can be precomputed; thus, the update rule for the *x* variable is fast. The following equations must be solved for the elements on the delineated triangular mesh, which are tabulated with ordinal numbers and solved separately:(20)$$\begin{array}{c} z_{i}^{n+1}=\underset{z_{i}}{argmin}\left(U_{i}\left(z_{i}\right)+\frac{1}{2}W_{i}{\left(D_{i}x^{n+1}-z_{i}+u_{i}^{n}\right)}^{2}\right)\\ u_{i}^{n+1}=u_{i}^{n}+D_{i}x^{n+1}-z_{i}^{n+1} \end{array}$$
where *i* represents the element number, and $z_{i}$ is a vector containing the gradient deformation matrix associated with element *i*. We update the variables $z_{i}$ and $u_{i}$ associated with each element separately, then update the global vectors *z* and *u*, and update the position vector *x* through Equation (19). The strain energy of each element is calculated by $U_{i}=US_{i}$, where $S_{i}$ is the initial area of the element, and $US_{i}$ is the value of the strain energy density function used to measure the deformation. In Equation (20), $U_{i}$ is used to update the $z_{i}$ value of each element.

### 3.4. Relationship between Internal Breast Targets and Manipulation Points

The contact point between the surface of the breast and the suction cup was set as the manipulation point with the letter *C*, and the internal target point of the breast was set as letter *T*. The position of the target point was determined by the image system, and the boundary point to the target point position was set as the control point position after the target point position was determined. The equations for solving breast tissue deformation were established in the previous sections, but the relationship between manipulation point *C* and target point *T* has not been established. Because breast tissue deformation is a nonlinear relationship, the relationship between the manipulation point and target point cannot be solved directly. Therefore, it is possible to linearize the model and solve it for correspondence. Assuming that tissue deformation proceeds slowly at all moments under low secularity, that is, the internal and external forces are equal, the problem can be regarded as a quasi-static problem, and Equation (7) can be rewritten as:(21)$$\frac{\partial U}{\partial x_{t}}-\lambda =0$$
(22)$$\left[\begin{array}{c} \frac{\partial U}{\partial x_{t}}\\ \frac{\partial U}{\partial x_{n}} \end{array}\right]=0$$
where $x_{c}$, $x_{n}$, $x_{t}$ and are the control point displacements, remaining point displacements, and target point displacements, respectively, λ is $F_{ext}$, and $\frac{\partial U}{\partial x_{c}}$ is the same as $F_{int}$ in Equation (8).

Now, linearizing Equation (22) gives:(23)$$A\delta x_{c}+B\delta x_{n}+C\delta x_{t}=0$$
where ${\left.A≜\left[\begin{array}{c} \frac{\partial^{2}U}{\partial x_{c}\partial x_{t}}\\ \frac{\partial^{2}U}{\partial x_{c}\partial x_{n}} \end{array}\right]\right|}_{x0}\in R^{(2t+2n)\times 2c}$, ${\left.B≜\left[\begin{array}{c} \frac{\partial^{2}U}{\partial x_{n}\partial x_{t}}\\ \frac{\partial^{2}U}{\partial x_{n}\partial x_{n}} \end{array}\right]\right|}_{x0}\in R^{(2t+2n)\times 2n}$, ${\left.C≜\left[\begin{array}{c} \frac{\partial^{2}U}{\partial x_{c}\partial x_{c}}\\ \frac{\partial^{2}U}{\partial x_{c}\partial x_{n}} \end{array}\right]\right|}_{x0}\in R^{(2c+2n)\times 2c}$, the equilibrium under the $x_{0}$ quasi-static problem consists of $\left(x_{c},x_{n},x_{t}\right)$. Where c, t, and n in the upper right corner of *R* represent the number of control points, target points, and the number of points other than the control points and the target points, respectively.

The need to establish a relationship between $\delta x_{n}$ and $\delta x_{c}$ reduces Equation (23) to a relationship containing only $\delta x_{c}$ and $\delta x_{t}$,
(24)$$\delta x_{n}=J_{n}^{T}\delta x$$
(25)$$\delta x_{t}=J_{t}^{T}\delta x$$
where $J_{n}^{T}$ and $J_{t}^{T}$ are matrices consisting of zeros and ones, respectively, and δx represents the displacement of all points inside the breast tissue.

Combining Equations (23) and (24),
(26)$$\delta x_{n}=J_{n}^{T}{\left(J_{n}^{T}\right)}^{-1}\delta x_{t}$$

Taking Equation (26) into Equation (23), and eliminating $\delta x_{n}$ yields
(27)$$\delta x_{c}=-A^{-1}\left[BJ_{n}^{T}{\left(J_{n}^{T}\right)}^{-1}+C\right]\delta x_{t}$$

The Jacobi matrix of the system is therefore,
(28)$$J=-A^{-1}\left[BJ_{n}^{T}{\left(J_{n}^{T}\right)}^{-1}+C\right]$$

Equation (7) show the relationship between the control point $\delta x_{c}$ and the target point $\delta x_{t}$. A linear conversion relationship for the control and target points was established.

### 3.5. PID Controller

The Jacobian relationship between the displacement $\delta x_{c}$ of the manipulation point and the $\delta x_{t}$ of the target point is solved under quasi-static conditions, and the experimental data obtained by direct tissue manipulation may have large errors. To eliminate this error, it is necessary to develop a method that can continuously correct the final position of the target point by using feedback. In Section 3.4, the relationship between the control point displacement and the target point displacement was linearized. Due to the advantages of PID control’s fast convergence, good stability, and simple implementation, PID feedback control can be used to achieve tissue manipulation. In the process of manipulating breast tissue deformation, the relationship between the manipulation point and target point must satisfy the relationship in Equation (7). In this study, the actuator was set to push and pull the breast tissue in the linear direction, and the error signal between the desired target point position $y_{p}$ and the actual target point position $y_{p}^{\prime }$ was defined as
(29)$$e=y_{p}-y_{p}^{\prime }$$

The input was the desired target position $y_{p}$, which was planned in advance, and the target error signal was the desired target displacement, which was solved using the linear prediction model obtained in Section 3.4, to determine the corresponding desired displacement of the manipulation point.
(30)$$\delta x_{c}=Je$$

The $\delta x_{c}$ signal is transmitted to the actuator and acts on the breast tissue. At the same time, the displacement of the control point $\delta x_{c}$ is introduced into the tissue dynamics model obtained in Section 3.4 to solve the updated model. The new Jacobian matrix is used to update the linear prediction model and combined with image technology to confirm the actual target position and feedback to the system. The system re-solves the model until the error approaches zero. Figure 10 shows the speed control block diagram of the actuator displacement control, and its formula is as follows
(31)$$\mu (t)=K_{1}\delta x_{c}+K_{2}\int_{t_{0}}^{t}\delta x_{c}dt+K_{3}\frac{d\delta x_{c}}{dt}$$
where $K_{1}$, $K_{2}$, $K_{3}$ are the PID control gains.

## 4. Simulation and Experimental Results

This section uses breast prosthesis and in vitro tissue (pork) as experimental objects for experiments. Pork was chosen as the experimental subject for the in vitro tissue manipulation experiment because the biomechanical properties of pork are similar to those of breast tissue and can better simulate the response of breast tissue. This similarity makes pork an ideal alternative material for evaluating the performance of surgical tools and robotic manipulation [25]. This article only needed to verify the accuracy of the target displacement, so the breast prosthesis was fixed on the platform, and the ball screw was used to realize the needle insertion and suction cup push–pull function. In order to better observe the changes in the target inside the tissue, this experiment did not place the breast prosthesis and in vitro tissue prone, and the suction cup was omitted to fix the tissue and only push and pull the tissue. The experimental device consisted of the following:(1)Prosthesis model: we used a flat hemispherical silicone prosthesis to simulate breast tissue.(2)In vitro tissue: we used pork (purchased from local supermarkets/qualified for quarantine/meeting experimental needs) as the experimental subject for the experiment.(3)Ball screw: the ball screw (GGP dual-axis ball-screw slide linear module, three-phase stepper motors—nema23 (57 mm); T-type micro precision ball-screw slide module, three-phase stepper motors—nema11 (28 mm) used in this experiment for the needle insertion and suction cup pushing and pulling of tissue.(4)Target recognition: we measured the target position using a CCD camera and an image processing computer.(5)Position of needle (18G), suction cup, and internal target: we computed the positions of the needle tip, suction cup, and target from the camera image.

### 4.1. Simulation

In order to verify the feasibility of the theory in Section 3, the tissue manipulation method was first simulated using MATLAB (2019-b) software and the simulation results were visualized. First, the two upper and lower two targets were defined under a 2D rectangle. The 2D breast tissue was modeled and meshed using Matlab’s PDE toolbox, and the meshed 2D breast tissue was considered as a rectangle in this section. Programming was performed using MATLAB for the computation on a Windows 10 computer with an Intel(R) Core(TM) i3-9100F CPU @ 3.60 GHz and 16 GB of running memory. When the positions of the control and target points were confirmed, tissue deformation was simulated, and the visualization results are shown in Figure 11. Figure 11a shows that when the target and other important organs were located inside a two-dimensional rectangular tissue, the manipulation point was displaced by applying a force perpendicular to the long side of the rectangle to pull the tissue downward, thereby allowing for a simple and ideal intervention path without obstacles. Figure 11b similarly demonstrates a simple intervention path by applying a force perpendicular to the long side of the rectangle to push the tissue downward. This simulation verified the displacement relationship between the manipulation point and the target pushed from Section 3 and achieved the manipulation of the target inside the tissue by pushing and pulling. To clearly describe the situation before and after tissue deformation, the needle path before undeformed tissue manipulation is represented as a purple rectangle, with the red point as the undeformed target point, and the blue semi-transparent matrix as the nerves, blood vessels, and other tissues inside the breast tissue. When the desired displacement position of the target was known, the desired displacement position of the target was entered to start tissue manipulation. The gray rectangle indicates the needle path after tissue manipulation and the green dots indicate the target position after tissue deformation. The results of this simulation clearly demonstrated that the tissue could be manipulated and deformed to displace the target at the desired position.

### 4.2. Breast Prosthesis and In Vitro Tissue Experiments

This process introduces how to use the CCD camera imaging principle to identify the target and control the position of the point. At present, this technology is mature. For a detailed derivation, please refer to the literature [26]. In the experiment, we used a CCD camera to verify the positions of the target point, manipulation point, and needle tip. The goal is to convert the world coordinates of any known point of the experimental object into pixel coordinates and, by initializing the camera parameters, obtain displacement data that are closer to the real value. Therefore, it is necessary to model the visual system of the CCD camera. Without considering the distortion of the CCD camera, the conversion relationship between points in the CCD camera coordinate system and points in the world coordinate system can be expressed as:(32)$$R_{c}=T_{cb}\cdot R_{m}$$
where $R_{c}={\left[X_{c},Y_{c},Z_{c},1\right]}^{T}$ is the homogeneous form of any point in the camera coordinate system, $T_{cb}$ is the rotation matrix and translation matrix from the world coordinate system to the camera coordinate system, and $R_{m}={\left[R_{x},R_{y},R_{z},1\right]}^{T}$ is the homogeneous form of any point in the world coordinate system.

Figure 12 is a schematic diagram of CCD camera imaging. According to the principle of similar triangles, we obtain:(33)$$\frac{AB}{oC}=\frac{AO_{c}}{oO_{c}}=\frac{RB}{rC}=\frac{X_{c}}{x}=\frac{Z_{c}}{f}=\frac{Y_{c}}{y}$$
(34)$$x=f\frac{X_{c}}{Z_{c}},y=f\frac{Y_{c}}{Z_{c}}$$
(35)$$Z_{c}\left[\begin{array}{c} x\\ y\\ 1 \end{array}\right]=\left[\begin{array}{cccc} f & 0 & 0 & 0\\ 0 & f & 0 & 0\\ 0 & 0 & 1 & 0 \end{array}\right]\left[\begin{array}{c} x_{c}^{f}\\ y_{c}^{f}\\ z_{c}^{f}\\ 1 \end{array}\right]$$
where f is the focal length, $Z_{c}$ is the scale factor, $x_{c}^{f}$, $y_{c}^{f}$, $z_{c}^{f}$ are the homogeneous coordinates of the space point in the camera coordinates and also the homogeneous coordinates of the image point in the image coordinate system.

Let the coordinates of the image projection point in the pixel coordinate system be $p={\left[u,v\right]}^{T}$, and let the coordinates of the image projection point in the image coordinate system be $p={\left[x,y\right]}^{T}$, then:(36)$$\left[\begin{array}{l} u\\ v\\ 1 \end{array}\right]=\left[\begin{array}{ccc} \frac{1}{dx} & 0 & u_{0}\\ 0 & \frac{1}{dy} & v_{0}\\ 0 & 0 & 1 \end{array}\right]\left[\begin{array}{l} x\\ y\\ 1 \end{array}\right]$$

In the formula, dx and dy are the pixel sizes, $u_{0}$ and $v_{0}$ are the image centers. Combining them, we obtain:(37)$$z_{c}\left[\begin{array}{c} u\\ v\\ 1 \end{array}\right]=\Omega \left[\begin{array}{c} x_{c}^{f}\\ y_{c}^{f}\\ z_{c}^{f}\\ 1 \end{array}\right]$$
where Ω is the internal parameter of the camera; $\Omega =\left[\begin{array}{cccc} f & 0 & u_{0} & 0\\ 0 & f & v_{0} & 0\\ 0 & 0 & 1 & 0 \end{array}\right]$.

Combining (32) and (37), we obtain:(38)$$z_{c}\left[\begin{array}{c} u\\ v\\ 1 \end{array}\right]=\Omega \cdot T_{cb}\left[\begin{array}{c} R_{x}\\ R_{y}\\ R_{z}\\ 1 \end{array}\right]$$

Based on the above relationships, the coordinates of any point in space can be converted to its pixel coordinates. In subsequent experiments, we used a CCD camera as the visual system to observe the displacement of the target point, manipulation point, and needle tip. By establishing a complete visual system model, we could monitor and provide real-time feedback on the position changes in the target and manipulation points during breast prosthesis experiments. In in vitro tissue experiments, we could infer the changes in the internal target point by observing the position changes in surface markers. Before manipulating the tissue, the positions of the manipulation point and the target point needed to be determined. During the manipulation process, the displacement of the target and manipulation points needed to be monitored and fed back in real-time. After the manipulation was completed, the deviation between the final target position and the expected position was determined, and the deviation between the needle tip and the target was calculated.

Subsequently, we conducted experiments on the breast prosthesis and in vitro tissue to verify that the error between the final actual position of the target point and the desired position was within the allowable range. The breast prosthesis needed for the experiments consisted of artificial silicone, which was chosen to allow the observation of the target point movement, and the elastic modulus of the breast prosthesis was determined to be 10 kPa [27]. The in vitro tissue required for the experiment was obtained from pork. A tissue manipulation experimental platform was mounted, and the breast prosthesis and in vitro tissues were fixed on the platform using a ball screw to drive the suction cup to manipulate the breast prosthesis and in vitro tissues, and another ball screw to drive the interventional needle to realize the puncture of the target point. The transparent prosthesis was fixed on the platform, and blue and black objects were placed inside the prosthesis, with the black object simulating the internal target site and the blue object simulating the vital tissue. The blue circles marked the desired target position, and the red circles marked the actual target position. The error between the deformed target position and desired target position was calculated by pushing the breast prosthesis through an external actuator. Similarly, the in vitro tissue was fixed on the platform to calibrate the desired target point position, and the target point was placed inside the in vitro tissue using a peg, with the peg inside the in vitro tissue and the head of the peg on the surface of the in vitro tissue to observe the displacement. Experiments were conducted on the breast prosthesis and in vitro tissue separately, and a CCD camera was used to observe the target point displacements in different experiments, as shown in Figure 13.

Finally, the actual results in the *X* and *Y* directions for the breast prosthesis and in vitro tissue were recorded each time, and a total of four targets were selected. Thirty experiments were performed for each target, and the statistical data were used to create target error maps, as shown in Figure 14. The box represents the 25th and 75th percentiles, the black horizontal line is the median, the solid triangle is the mean, and the whiskers represent the maximum and minimum values. In the breast prosthesis experiment, the error in the *X* direction perpendicular to the force direction of the two targets was small, with a maximum value of 0.7 mm, and the maximum error in the *Y* direction was 1.7 mm. This error was in accordance with the permissible target localization error in interventional breast surgery; that is, this study’s research content provided a research method for interventional breast surgery. In the in vitro tissue experiments, the error in the *X* direction perpendicular to the force direction of the two targets was smaller, with a maximum value of 1.3 mm, whereas the maximum error in the *Y* direction was 2.5 mm. The main factors leading to the error in this research plan were that in order to clearly observe the experimental results, the breast prosthesis and in vitro tissue were not placed prone, and the vertical mobility was not restricted when manipulating the tissue. This error was in accordance with the target localization error allowed in breast interventional surgery [27]; that is, the research content of this study provided a research method for breast interventional surgery.


## 5. Summary and Outlook

This paper proposed a breast interventional robot that mimicked the scorpion’s hunting process. Through tissue manipulation, the target was ultimately shifted to a simple and safe puncture path, improving the accuracy of interventional surgery. This robot realizes a multidisciplinary cross-integration innovation, and its main features are as follows:Bionic design: It was proposed to use a suction cup to absorb breast tissue and manipulate the deformation of breast soft tissue by pushing and pulling to shift the target to a simple and safe puncture path for needle insertion. The suction cup absorbed the tissue to avoid damage caused by traditional steel plate pressing. The robot’s workflow imitated the process of scorpion hunting, allowing the robot’s puncture module and tissue manipulation module to work together to perform interventional surgery.Real-time deformation of breast tissue: Since the robot tissue manipulation module limited the fluidity of breast tissue in the vertical direction, a new method was proposed on the basis of ignoring the influence of tissue force in the vertical direction, which regarded the tissue as composed of multiple superimposed planes. The manipulation point of the breast tissue and the target point inside the breast tissue could always be located in one of the multiple planes at the same time. Therefore, the deformation problem of three-dimensional breast tissue was simplified to the problem of two-dimensional breast tissue deformation. The ADMM algorithm could quickly solve the deformation of breast tissue, ensuring the real-time control of breast deformation during interventional surgery.Accuracy of target manipulation: The problem of breast tissue deformation under low-speed manipulation was regarded as a quasi-static breast tissue deformation problem. The linear Jacobian relationship between the manipulation point and the target was derived. PID closed-loop control was used to reduce the error of the manipulation target displacement. The maximum positioning error obtained in the experiment was 2.5 mm.

The current limitations of this approach are as follows:In this study, we used CCD cameras instead of medical imaging systems, which could not accurately locate the target points inside the tissue when conducting experiments on in vitro tissues.After the interventional needle enters the tissue, the needle and tissue interact and deform, but this article ignored the deformation caused by the interaction between the needle and tissue.

In the future work, we will work under the guidance of a medical imaging system, considering the deformation of the needle and tissue, and study multi-point tissue manipulation in three-dimensional conditions by training models to obtain a large quantity of data from various points inside the breast tissue under different deformations, using machine learning to train the breast tissue deformation data, establishing the relationship between the control points and target points in three dimensions, and combining a nonlinear controller to compensate for the error to achieve the manipulation.

## Figures and Tables

**Figure 1 sensors-24-06408-f001:**
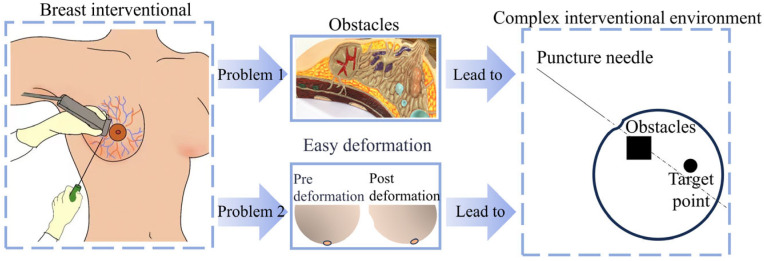
Complex breast interventional procedure environment.

**Figure 2 sensors-24-06408-f002:**
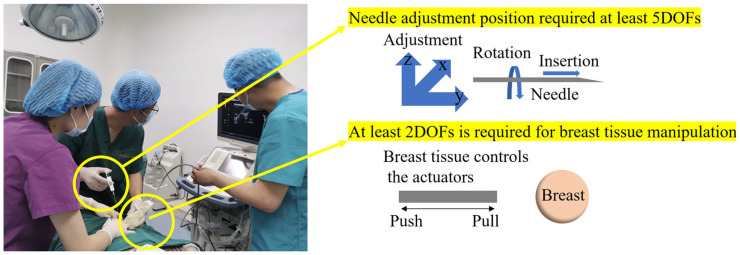
Required DOFs for the puncture module and tissue manipulation module.

**Figure 3 sensors-24-06408-f003:**
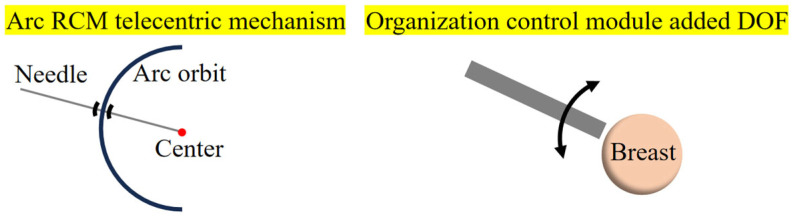
New DOFs’ diagram example.

**Figure 4 sensors-24-06408-f004:**
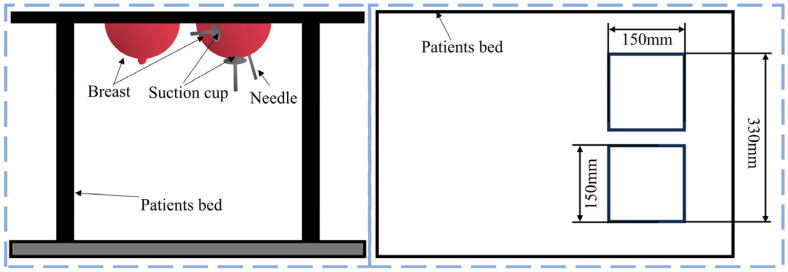
Schematic diagram of robot working and size design.

**Figure 5 sensors-24-06408-f005:**
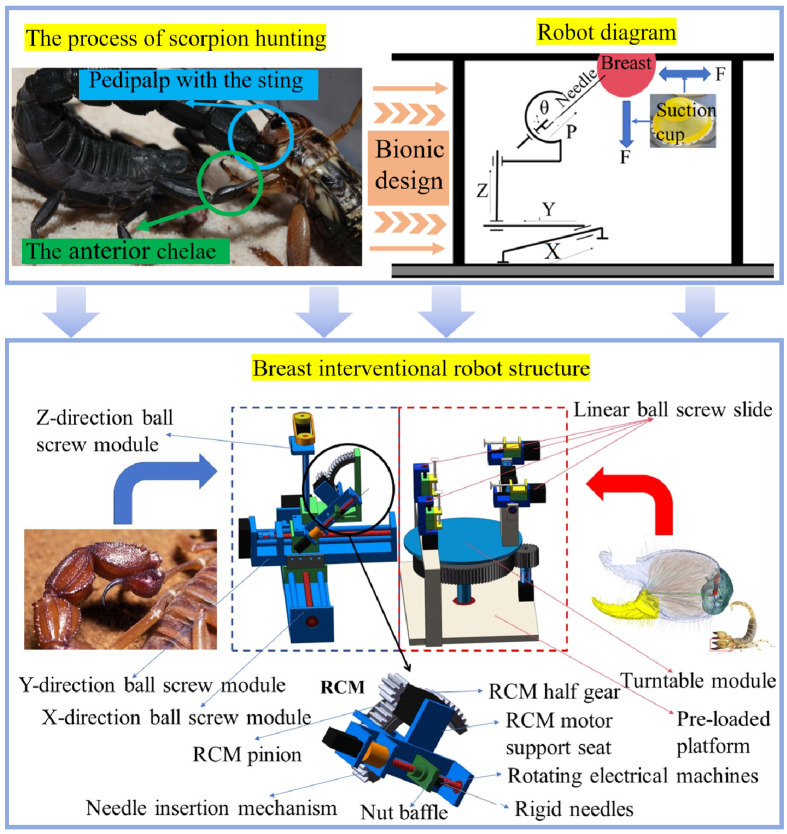
Bionic concept and design.

**Figure 6 sensors-24-06408-f006:**
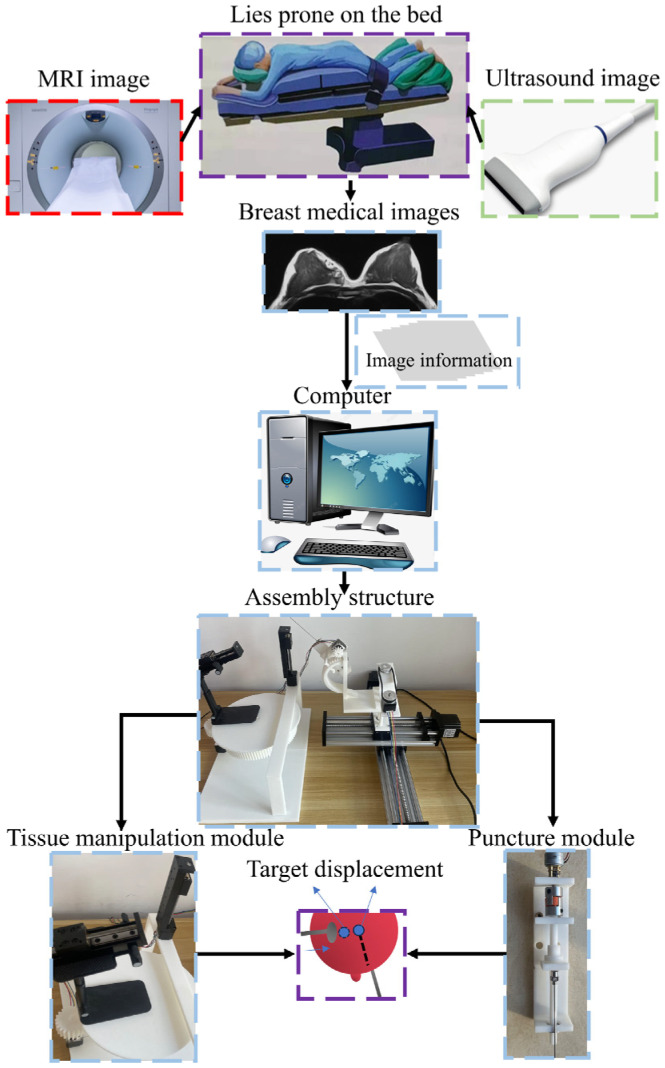
Overall system diagram.

**Figure 7 sensors-24-06408-f007:**
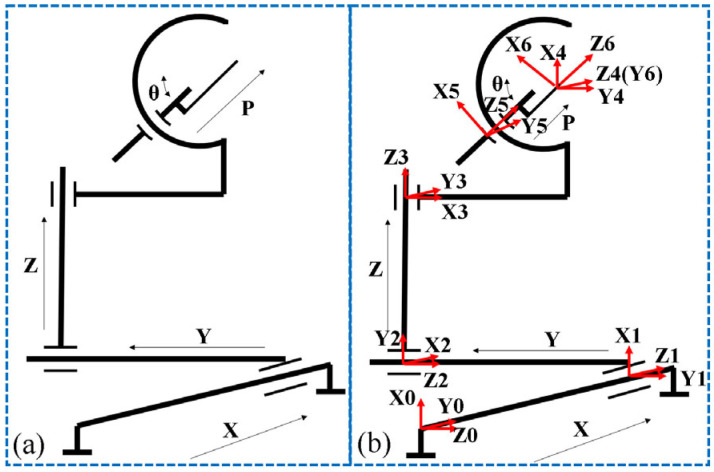
Sketch of the puncture module model and the corresponding joint reference coordinate system. Where, subfigures (**a**) is the sketch of the puncture module model, and subfigures (**b**) is the joint coordinate diagram of the puncture module.

**Figure 8 sensors-24-06408-f008:**
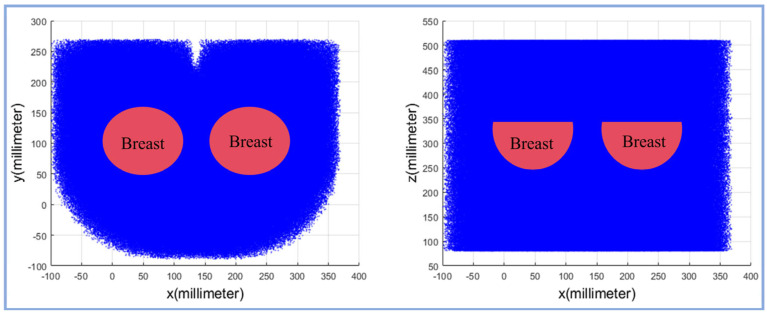
Robot workspace.

**Figure 9 sensors-24-06408-f009:**
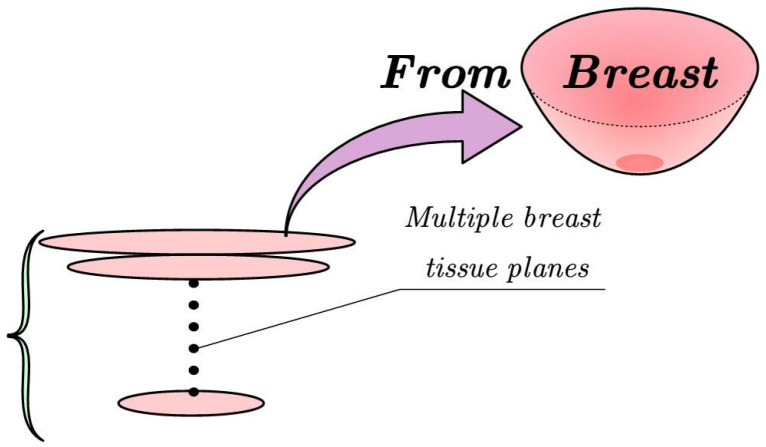
Breast tissue stratification study.

**Figure 10 sensors-24-06408-f010:**
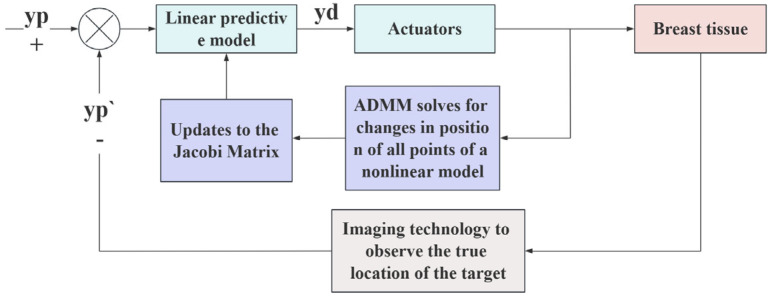
PID controller block diagram.

**Figure 11 sensors-24-06408-f011:**
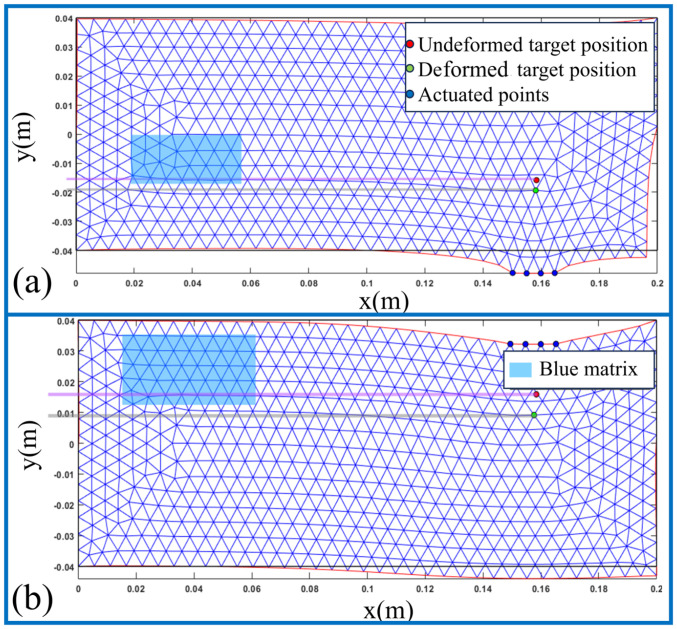
Simulation visualization of realized push–pull tissue deformation, where subfigures (**a**) is simulation visualization of realized pull tissue deformation, subfigures (**b**) for Simulation visualization of realized push tissue deformation.

**Figure 12 sensors-24-06408-f012:**
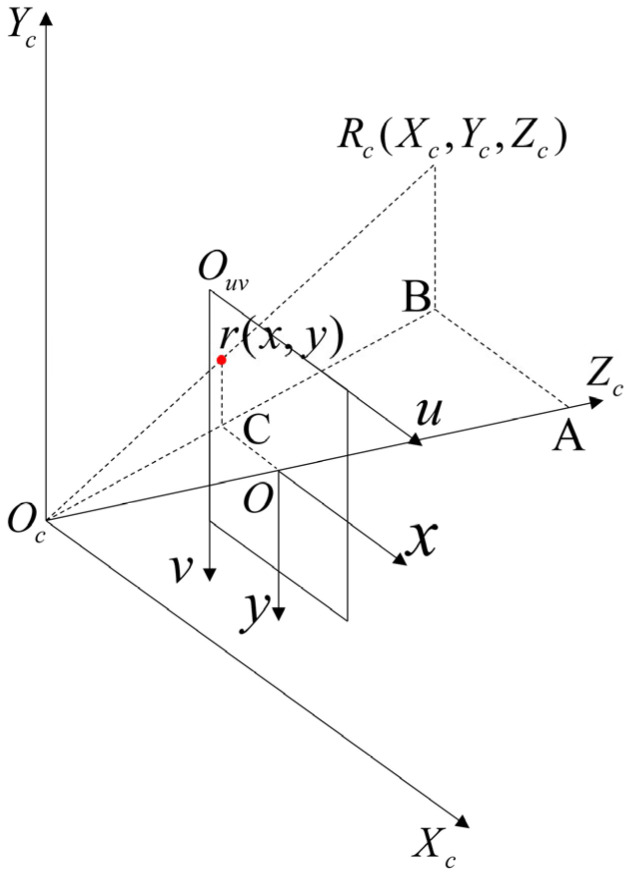
Schematic diagram of CCD camera imaging.

**Figure 13 sensors-24-06408-f013:**
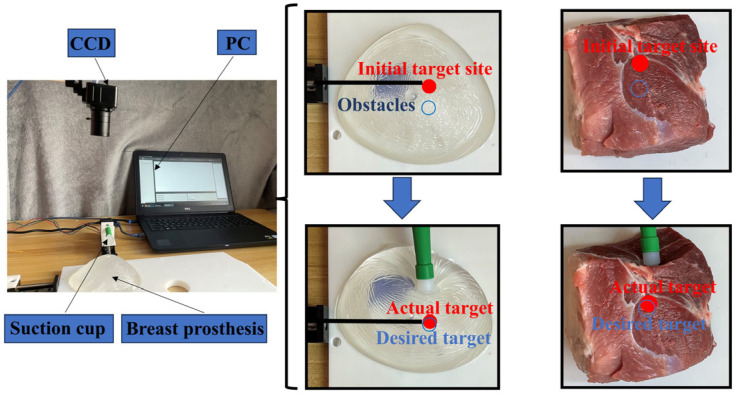
Tissue manipulation target displacement experiment.

**Figure 14 sensors-24-06408-f014:**
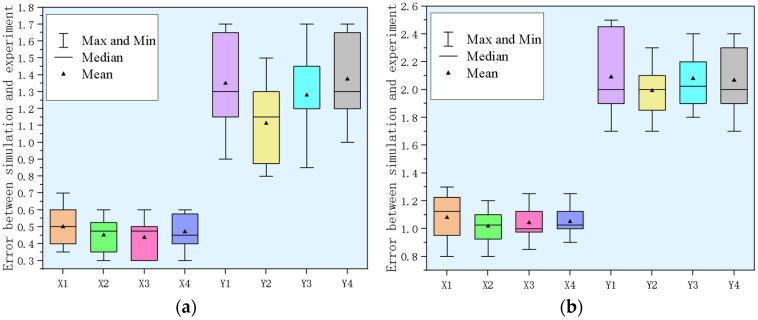
Experimental target localization error box plot: (**a**) Box plot of breast prosthesis target localization error; (**b**) box plot of target localization error in in vitro tissue.

**Table 1 sensors-24-06408-t001:** DH method parameter meaning.

Notation	Concrete Meaning
$a_{i-1}$	$\mathrm{Along}\;\mathrm{the}\;X_{i-1}\text{-}\mathrm{direction},\;\mathrm{length}\;\mathrm{of}\;\mathrm{the}\;\mathrm{common}\;\mathrm{perpendicular}\;\mathrm{between}\;Z_{i-1}\;\mathrm{and}\;Z_{i}$
$\alpha_{i-1}$	$\mathrm{Along}\;\mathrm{the}\;X_{i-1}\text{-}\mathrm{direction},\;\mathrm{angle}\;\mathrm{of}\;\mathrm{rotation}\;\mathrm{of}\;Z_{i-1}\;\mathrm{to}\;Z_{i}$
$d_{i}$	$\mathrm{Along}\;\mathrm{the}\;Z_{i}\text{-}\mathrm{direction},\;\mathrm{length}\;\mathrm{of}\;\mathrm{the}\;\mathrm{common}\;\mathrm{perpendicular}\;\mathrm{between}\;X_{i-1}\;\mathrm{and}\;X_{i}$
$\theta_{i}$	$\mathrm{Along}\;\mathrm{the}\;Z_{i}\text{-}\mathrm{direction},\;\mathrm{angle}\;\mathrm{of}\;\mathrm{rotation}\;\mathrm{of}\;X_{i-1}\;\mathrm{to}\;X_{i}$

**Table 2 sensors-24-06408-t002:** Table of DH parameters.

Joints	$\alpha_{i-1}$	$a_{i-1}$	$\theta_{i}$	$d_{i}$
1	0°	0	0°	$d_{1}$ (0–430_mm)
2	−90°	$a_{1}$ (60_mm)	−90°	$d_{2}$ (0–200_mm)
3	−90°	0	−90°	$d_{3}$ (0–150_mm)
4	−90°	$a_{3}$ (70_mm)	$\theta_{4}$(−135°~45°)	0
5	−90°	0	$\theta_{5}$(−90°)	$d_{5}$ (0–80_mm)
6	0°	$a_{5}$ (80_mm)	0°	$d_{6}$ (80_mm)

## Data Availability

The data that support the findings of this study are available from the corresponding author upon reasonable request.

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
