# Peer review of "Study on Bionic Design and Tissue Manipulation of Breast Interventional Robot"

_sensors, 2024, doi:10.3390/s24196408_

Round 1

Reviewer 1 Report

Comments and Suggestions for Authors

The authors introduce a robot design for the breast manipulation and intervention. The paper first considers the robot kinematics and then proposes the breast deformation model. The experiments illustrate the developed techniques. Although the authors’ idea looks mostly clear for me, unfortunately, the paper presentation is very weak: there are many inaccurate computations, unexplained notations, and unconventional terms, which make it challenging to understand, appreciate, and reproduce the authors’ work. In addition, I am not sure that the robot design and its kinematics will be suitable for the considered operations. Please check the comments below for the details.

Major comments:

1.        In Subsection 2.1, it is difficult to imagine the desired DOFs of the robot. If possible, I recommend the authors provide a relevant figure.

2.        Similar to the previous comment, it is also challenging to comprehend the material given in the paragraph above Fig. 2, where the authors again discuss the robot DOFs.

3.        The proposed robot includes four prismatic joints. It is known that the four prismatic joints are “linearly dependent.” In other words, if we fix the output link of the robot, there will be an infinite number of solutions to the inverse kinematic problem. On the other hand, the output link of the considered robot will actually have four DOFs instead of five (ignoring the rotation about the tool axis) because of the internal DOF. In this regard, is becomes unclear if the proposed robot meets the required DOFs.

4.        The kinematic analysis presented on p. 9 is challenging to understand, for example:

4.1.       It is unclear how the “telecentric localization mechanism” (see also comment no. 7) is mainly adjusted.

4.2.       Equation (5) looks incorrect and meaningless.

4.3.       Figure 6 illustrates the robot workspace, but the authors do not mention for which orientation of the output link they derived this workspace. Is it a maximum workspace (for any orientation)?

5.        The model of the breast tissue presented in Subsections 3.2–3.4 looks inaccurate as well:

5.1.       P. 11, l. 328. It is unclear what the authors mean under “breast tissue is closely related to the time response.”

5.2.       It is unclear how Eqs. (12)–(14) are related to other equations and material of the paper.

5.3.       Eq. (14). How do the authors compute this partial derivative if C is a tensor according to p. 11, l. 335?

5.4.       Eq. (18). What is z here?

5.5.       Eqs. (19)–(20). The style of notations has changed, and it becomes challenging to comprehend these equations.

5.6.       P. 13, l. 378. It is unclear why the authors return to Eq. (7) here.

5.7.       P. 14, l. 383–384. Parameters 2t, 2n, and 2c appear without explanations, as well as subscript p in the expression of matrix B.

5.8.       Eqs. (27) and (28). What is Jc?

6.        The computations presented in Subsection 4.2 (Eqs. (32)–(38)) are also difficult to comprehend, because it is unclear what is the given data and what the authors should calculate. In addition, the equations have many unexplained notations whose style also changes between the equations. The authors must carefully revise this material.

7.        There are many typos or strange and unclear terms and phrases that should be revised, for example, “robot breast interventional robot” (p. 2, l. 61), “improvement of D-H method” (p. 6, l. 219), “moving sub” (p. 7, l. 233), “invariant matrix,” “variation matrix” (p. 8, l. 247–248), “telecentric localization mechanism of the curved orbit” (p. 9, l. 252–253), “arc deflection joint”, “translational puncture joint” (p. 9, l. 256), “change matrix” (p. 9, l. 259), “solution of… deformation,” “solve… deformation” (p. 10, l. 277–279), “Eulerian coordinate system” (p. 10, l. 296), “continuum medium mechanics” (p. 10, l. 299), “Lunger-Kutta” (p. 11, l. 313), “Considering that the gradient” (p. 11, l. 320), “trapezoidal matrix space” (p. 12, l. 359), “matrices consisting of zero and one,” “displacement matrix” (p. 14, l. 388–389), “linear prediction model” (p. 14, l. 394), “stratification study” (p. 15, l. 421), “57/28 stepper motor” (p. 15, l. 434–435), “MATALB” (p. 16, l. 443), “simplifying the intervention path” (p. 16, l. 453), “defommed” (p. 16, Fig. 9a).

Minor comments:

8.        P. 2, l. 51. The authors write “Lu et al.” and cite paper [6], but the authors of this paper are Du and Liu.

9.        P. 6, Table 1. There should be αi–1 instead of ai–1 in the second row of the table.

10.    P. 7, l. 223. I recommend the authors provide a relevant reference for the modified D-H method.

11.    P. 7, l. 232. What is the range of parameter i?

12.    P. 7, l. 232–233. The authors write that the direction of Zi–1 is collinear with the corresponding joint axis or displacement direction. This statement contradicts the direction of Z0 in Fig. 5b. I recommend the authors check the directions of other coordinate axes in the figure as well.

13.    P. 7, Table 2. The authors do not specify the units of parameters ai–1 and di.

14.    P. 8, l. 247. Shouldn’t there be 65T instead of 10T?

15.    P. 8, Eq. (3). I recommend the authors use a consistent italic style of the scalar variables in the matrix.

16.    P. 8, the equations under l. 250. Operator “.*” looks unconventional.

17.    P. 9, l. 261. Matrix notation 63T looks inappropriate here.

18.    P. 9, Eq. (6). The dot notations in the last column look unclear.

19.    P. 9, l. 268. The authors refer to the black lines, but Fig. 6 has no such lines.

20.    P. 10, l. 297. The authors write M denotes the mass matrix, but this notation denotes mass according to l. 290.

21.    P. 10, l. 305. The authors write, “we must solve Eq. (7),” but it is unclear which variable the authors must solve this equation for.

22.    P. 11, Eq. (9):

22.1.   Parameter Δt appears unexplained.

22.2.   Why does function f depend on t instead of t + Δt in the second line of the equation?

22.3.   Why does parameter Fext depend on t and parameter Fint depend on t + Δt in the third line of the equation?

23.    P. 11, l. 308. Superscript d appears without explanations.

24.    P. 11, l. 314. It is unclear why the conventional numerical methods are not suitable for solving the equations of the breast model.

25.    P. 11, l. 324. The authors introduce abbreviation ADMM but do not spell it out explicitly.

26.    P. 12, l. 342. Parameter J appears unexplained.

27.    P. 12, l. 345. It is unclear what the authors mean under parameter I4.

28.    P. 12, l. 360. The authors write W is the weighting matrix, but they have already used this notation in Eq. (12).

29.    P. 13, l. 363. It is unclear which matrix in Eq. (19) the authors refer to.

30.    P. 15, l. 412–417. The sentence given in these lines looks too challenging to read, and I recommend the authors rewrite it.

31.    P. 15, l. 418. It is unclear what actuator the authors consider here.

32.    P. 15, Fig. 8. The figure contains unexplained notations yp and yd, and notation yp` appears above two signal lines, which looks incorrect.

Comments on the Quality of English Language

There are many typos or strange and unclear terms and phrases that should be revised (see comment no. 7).

Author Response

Comments 1: In Subsection 2.1, it is difficult to imagine the desired DOFs of the robot. If possible, I recommend the authors provide a relevant figure.

Response 1: Thank you for pointing this out. We agree with this comment. Therefore, we have revised Section 2.1 to further clarify the robot degrees of freedom (DOF). Specifically, we now explain that in clinical practice, surgeons usually use one hand to manipulate breast tissue to adjust its deformation, while the other hand adjusts the position and angle of the puncture needle for operation. Based on this clinical approach, we designed the breast intervention robot into two modules: the puncture module and the tissue control module, without the integration of the medical imaging system. In addition, we added Figure 2 to help readers more clearly understand the key role of degrees of freedom in robot operation. These revisions can be found on page 3, from lines 128–147, and on page 4, line 155.

Comments 2: Similar to the previous comment, it is also challenging to comprehend the material given in the paragraph above Fig. 2, where the authors again discuss the robot DOFs.

Response 2: Thank you for pointing this out. We agree with this comment. Therefore, we have further explained in detail the working principle of the arc remote center of motion (RCM) mechanism and explicitly pointed out that the arc RCM mechanism provides an additional 1 DOF to the system. At the same time, we explained the purpose of adding 1 DOF to the tissue control module to enhance the ability to manipulate the tissue. In addition, we added Figure 3 to help readers more intuitively understand the basic principle of the increased degree of freedom of this module. For more details about the RCM mechanism, please refer to reference [17]. These revisions can be found on page 4, from lines 164 and 170–188.

Comments 3: The proposed robot includes four prismatic joints. It is known that the four prismatic joints are “linearly dependent.” In other words, if we fix the output link of the robot, there will be an infinite number of solutions to the inverse kinematic problem. On the other hand, the output link of the considered robot will actually have four DOFs instead of five (ignoring the rotation about the tool axis) because of the internal DOF. In this regard, is becomes unclear if the proposed robot meets the required DOFs.

Response 3: Thank you for pointing this out. Regarding the reviewer's comment on the inverse kinematics problem, we acknowledge that there is indeed the possibility of multiple solutions. However, by fixing the first three joints, the solution can be made unique. The four prismatic joints of the proposed robot are "linearly dependent", so we fixed the first three joints when solving the inverse kinematics. The insertion freedom of the needle cannot be ignored, so 3 DOFs are provided in the x, y, and z directions, 1 DOF is provided by the arc RCM mechanism, and 1 DOF is provided by the needle insertion drive, for a total of 5 DOFs (ignoring the rotation of the needle around its own axis). The positioning accuracy of the needle tip is the core requirement of the system. We also re-emphasized this point in Section 2.1 to ensure consistency with the previous revisions to the first two comments to avoid misunderstandings by readers. Thank you for your valuable comments.

Comments 4: The kinematic analysis presented on p. 9 is challenging to understand, for example:

4.1.       It is unclear how the “telecentric localization mechanism” (see also comment no. 7) is mainly adjusted.

4.2.       Equation (5) looks incorrect and meaningless.

4.3.       Figure 6 illustrates the robot workspace, but the authors do not mention for which orientation of the output link they derived this workspace. Is it a maximum workspace (for any orientation)?

Response 4: Thank you for pointing this out. We agree with this comment.

4.1      In the reply to question 2, we have added an explanation of the telecentric mechanism and attached reference [17]. We will address the unclear words and expressions while solving question 7 later. If you want to understand the telecentric positioning mechanism, you can read reference [17].

4.2       Since the robot has four linearly dependent prismatic joints, there will be multiple solutions when solving the inverse kinematics analysis, and the analysis is complicated. Therefore, the inverse kinematics analysis process is simplified and only joints 4, 5, and 6 are solved. After fixing joints 1, 2, and 3, 30T is a constant. Therefore, we transform the position of the end joint in the base coordinate  (px,py,pz) into the position of the end joint relative to the locked joint coordinate (p'x,p'y,p'z). Equation (5) is a standard form based on homogeneous coordinate transformation, which is used to describe the needle tip position of the puncture module. Although the matrix form is more complicated, it does conform to the transformation relationship in robot kinematics and is used to accurately control the needle tip position.

4.3        Regarding the description of Figure 6, Figure 6 shows the working space of the needle tip of the puncture module. This may be because the previous expression was not clear enough, causing to misunderstand that it is the working space at the end of the robot, but it is actually the working space of the needle tip. This working space covers the entire operating range required for the surgery, including breast tissue in different locations. Even when the puncture module is at the farthest end, the needle tip can still accurately puncture the target tissue. (The updated document Figure 6 becomes Figure 8.)These revisions can be found on page 10, from lines 289–298.

Comments 5: The model of the breast tissue presented in Subsections 3.2–3.4 looks inaccurate as well:

5.1.       P. 11, l. 328. It is unclear what the authors mean under “breast tissue is closely related to the time response.”

5.2.       It is unclear how Eqs. (12)–(14) are related to other equations and material of the paper.

5.3.       Eq. (14). How do the authors compute this partial derivative if C is a tensor according to p. 11, l. 335?

5.4.       Eq. (18). What is z here?

5.5.       Eqs. (19)–(20). The style of notations has changed, and it becomes challenging to comprehend these equations.

5.6.       P. 13, l. 378. It is unclear why the authors return to Eq. (7) here.

5.7.       P. 14, l. 383–384. Parameters 2t, 2n, and 2c appear without explanations, as well as subscript p in the expression of matrix B.

5.8.       Eqs. (27) and (28). What is Jc?

Response 5: Thank you for pointing this out. We agree with this comment.

5.1     Breast tissue has significant nonlinearity and anisotropy (different physical properties in different directions), and its deformation occurs gradually over time after being stressed, which shows that its deformation behavior is closely related to time and exhibits viscoelastic properties. . Therefore, a simple mass-spring damping model cannot accurately describe its deformation, and a hyperelastic model must be used to better simulate its stress-strain relationship (stress is the applied force and deformation is the resulting deformation). These revisions can be found on page 12, from lines 356–362.

5.2       Equations (12)–(14) describe the deformation mechanics of breast tissue using the Cauchy-Green deformation tensor and the second Piola-Kirchhoff stress tensor. These equations are fundamental to modeling the nonlinear elastic response of tissue, which is necessary to accurately predict the effects of manipulation during interventional procedures.

Equations (12)–(14) form the basis for the strain energy function used in the deformation model described in Equation (7). These equations are essential for calculating the internal forces acting on the breast tissue, which are then incorporated into the overall dynamic model of tissue deformation.

The stress-strain relationship established by equations (12)-(14) enables us to simulate the response of breast tissue under external manipulation, which is crucial for predicting tissue behavior during minimally invasive surgery. This relationship has a direct impact on the tissue manipulation algorithm described in Section 4.

5.3         The tensor C in equation (14) refers to the right Cauchy-Green deformation tensor, defined as , where F is the deformation gradient matrix. The partial derivative  of the strain energy function G with respect to the tensor C is calculated based on the selected hyperelastic material model, such as the NeoHookean model. In this model, the chain rule is used to differentiate the strain energy function to obtain the expression of the second Piola-Kirchhoff stress tensor S. Since this is a standard hyperelastic material model, the specific partial derivative calculation process has been described in detail in the relevant literature, and this study is calculated according to this standard method.Related calculation references:

Bonet, J., & Wood, R.D. (1997). Nonlinear Continuum Mechanics for Finite Element Analysis. Cambridge University Press.

Holzapfel, G.A. (2000). Nonlinear Solid Mechanics: A Continuum Approach for Engineering. Wiley.

5.4        In the equation, z=Dx, but in the article, Z is capitalized, which caused ambiguity and has been corrected. This is the common step of the ADMM algorithm. These revisions can be found on page 13, from lines 390–391.

5.5      Thank you for your valuable comments on the change of notation style in equations (19)–(20). We have modified these equations to ensure that the notation is consistent with the rest of this paper. Given that equations (19) and (20) involve relatively complex mathematical derivations, understanding these notation changes may increase the difficulty of understanding. To improve the readability and understandability of the equations, we will further explain their relationship with the content of Section 3.2. In addition, considering the complexity of the derivation of the equations, we also provide relevant references [20] so that readers can have a deeper understanding of the source and background of the equations. These revisions can be found on page 14, from lines 398–404.

5.6        We cite equation (7) for the purpose of deriving a linear relationship of the system under quasi-static conditions in this section. In the subsequent derivation, the displacement relationship between the target point and the control point is also based on the equilibrium equation of internal and external forces under quasi-static conditions. This can be understood as processing nonlinear changes by linearization to simplify the analysis.

5.7        The subscript p in the expression of the matrix B has been corrected. The symbols 2t, 2n, and 2c denote the target points, non-target points, and additional points of the control points, respectively. The number of control points can be indicated in the text by the rows and columns of the matrix. These revisions can be found on page 15, from lines 424–426.

5.8       This is a typo. Jc has been changed to Jt. These revisions can be found on page 15, from Eqs(27-28).

Comments 6: The computations presented in Subsection 4.2 (Eqs. (32)–(38)) are also difficult to comprehend, because it is unclear what is the given data and what the authors should calculate. In addition, the equations have many unexplained notations whose style also changes between the equations. The authors must carefully revise this material.

Response 6: Thank you for pointing this out. We agree with this comment. Section 4.2 introduces the basic principle of CCD camera imaging, which is now a relatively mature technology. We have added a description of the simple imaging principle and function of CCD cameras. We did find that there are unexplained symbols in the equations and the style is not uniform, so we have unified the symbols and style. These revisions can be found on line 517-519, from Eqs(35,37).

Comments 7: There are many typos or strange and unclear terms and phrases that should be revised, for example, “robot breast interventional robot” (p. 2, l. 61), “improvement of D-H method” (p. 6, l. 219), “moving sub” (p. 7, l. 233), “invariant matrix,” “variation matrix” (p. 8, l. 247–248), “telecentric localization mechanism of the curved orbit” (p. 9, l. 252–253), “arc deflection joint”, “translational puncture joint” (p. 9, l. 256), “change matrix” (p. 9, l. 259), “solution of… deformation,” “solve… deformation” (p. 10, l. 277–279), “Eulerian coordinate system” (p. 10, l. 296), “continuum medium mechanics” (p. 10, l. 299), “Lunger-Kutta” (p. 11, l. 313), “Considering that the gradient” (p. 11, l. 320), “trapezoidal matrix space” (p. 12, l. 359), “matrices consisting of zero and one,” “displacement matrix” (p. 14, l. 388–389), “linear prediction model” (p. 14, l. 394), “stratification study” (p. 15, l. 421), “57/28 stepper motor” (p. 15, l. 434–435), “MATALB” (p. 16, l. 443), “simplifying the intervention path” (p. 16, l. 453), “defommed” (p. 16, Fig. 9a).

Response 7: Thank you for pointing this out. We agree with this comment. Therefore, we have reexpressed it according to your request and marked it in red to highlight it. For example,breast interventional robot(l.61),DH method parameter meaning(l.242),moving sub is deleted,fix matrix,matrix(l.269),telecentric localization mechanism of the curved orbit The corresponding passageand“arc deflection joint”, “translational puncture joint” (p. 9, l. 256) has been reexpressed,(l.274-279),matrix(l.281),“solution of… deformation,” “solve… deformation”has been reexpressed(l.303-305),“Eulerian coordinate system”has been reexpressed(l.323),“continuum medium mechanics” (p. 10, l. 299)has been reexpressed(l.326),“Lunger-Kutta” (p. 11, l. 313)has been reexpressed(l.342),“trapezoidal matrix space” (p. 12, l. 359) has been reexpressed(l.392),“matrices consisting of zero and one,” “displacement matrix”has been reexpressed(l.430-431),“linear prediction model” has been reexpressed(l.436),the remaining revisions are in (l.480-482,490,500,fig11)

Comments 8: P. 2, l. 51. The authors write “Lu et al.” and cite paper [6], but the authors of this paper are Du and Liu.

Response 8: We have corrected “Lu et al.” to “Du and Liu” in line 51 on page 2 and ensured that the author information of the cited reference [6] is accurate. Thank you again for your careful correction. These revisions can be found on page 2, from line 51.

Comments 9: P. 6, Table 1. There should be αi–1 instead of ai–1 in the second row of the table.

Response 9: Thank you for pointing out the error in Table 1 . We have corrected the ai–1 in the second row to αi–1 and ensured that the symbols in the table are used correctly. Thank you again for your careful correction. These revisions can be found on page 2, from line 51.These revisions can be found on page 7, from line 242.

Comments 10: P. 7, l. 223. I recommend the authors provide a relevant reference for the modified D-H method.

Response 10: Thank you for pointing this out. We agree with this comment. Therefore, we have based on your comments, we have added relevant references for the improved D-H method to further support the content in the paper. These revisions can be found on page 7, from line 247.

Comments 11: P. 7, l. 232. What is the range of parameter i?

Response 11: Thank you for pointing this out. We agree with this comment. Therefore, we have clarified the range of parameter i, which is from 0 to 6. Thanks again for your careful correction. These revisions can be found on page 8, from line 155.

Comments 12: P. 7, l. 232. What is the range of parameter i?

Response 12: Thank you for pointing this out. We agree with this comment. Therefore, we have delete the sentence “the direction of Zi–1 is collinear with the corresponding joint axis or displacement direction.” This sentence is not clear enough. For more details, please refer to [19].These revisions can be found on page 8, from line 253-256.

Comments 13: P. 7, Table 2. The authors do not specify the units of parameters ai–1 and di.

Response 13: Thank you for pointing this out. We agree with this comment. Therefore, we have added the relevant unit information in the table. Thank you again for your valuable feedback. These revisions can be found on page 8, from line 262(Table2).

Comments 14:  P. 8, l. 247. Shouldn’t there be 65T instead of 10T?

Response 14: Thank you for pointing this out. We agree with this comment. Therefore, we have modified it to replace 10T with 65T. These revisions can be found on page 9, from line 269.

Comments 15: P. 8, Eq. (3). I recommend the authors use a consistent italic style of the scalar variables in the matrix.

Response 15: Thank you for pointing this out. We agree with this comment. Therefore, we have made scalar variables in matrices consistent with italic style. These revisions can be found on page 9, from line 271-272(Eq. (3)).

Comments 16: P. 8, the equations under l. 250. Operator “.*” looks unconventional.

Response 16: Thank you for pointing this out. We agree with this comment. Therefore, we have removed an irregular use of the ".*" operator. We have corrected it to ensure that the symbol is in accordance with common usage. These revisions can be found on page 9, from line 272-273.

Comments 17: P. 9, l. 261. Matrix notation 63T looks inappropriate here.

Response 17: Thank you for pointing this out. We think that 63T's expression here is correct because we want to calculate the inverse kinematics solution after fixing the first three joints. If 63T has other objections here, please contact us in time for modification. We will respect your opinions and make reasonable modifications as soon as possible.

Comments 18: P. 9, Eq. (6). The dot notations in the last column look unclear.

Response 18: Thank you for pointing this out. We agree with this comment. Therefore, we have revised it to ensure that the symbol expression is clearer and more standardized. These revisions can be found on page 9, from Eq. (6).

Comments 19: P. 9, l. 268. The authors refer to the black lines, but Fig. 6 has no such lines.

Response 19: Thank you for pointing this out. We agree with this comment. Therefore, we have deleted the bolded words and rewritten the entire paragraph to make it clearer. These revisions can be found on page 10, from line 289-298.

Comments 20: P. 10, l. 297. The authors write M denotes the mass matrix, but this notation denotes mass according to l. 290.

Response 20: Thank you for pointing this out. We agree with this comment. Therefore, we have remove ambiguous expressions and keep only the interpretation of the quality matrix. These revisions can be found on page 11, from line 316, 323.

Comments 21: P. 10, l. 305. The authors write, “we must solve Eq. (7),” but it is unclear which variable the authors must solve this equation for.

Response 21: Thank you for pointing this out. The main variable we need to solve in equation (7) here is the displacement vector x, which describes the displacement state of the system. Equation (7) describes the motion relationship of the system under the balance of internal and external forces, where x is the key variable we need to solve by numerical methods. In actual calculations, we use the implicit Euler integration method to solve the time step to ensure the stability of the system. Therefore, equation (7) can help us obtain the quasi-static solution of the system by solving the displacement vector x.

Comments 22:  P. 11, Eq. (9):

22.1.   Parameter Δt appears unexplained.

22.2.   Why does function f depend on t instead of t + Δt in the second line of the equation?

22.3.   Why does parameter Fext depend on t and parameter Fint depend on t + Δt in the third line of the equation?

Response 22:Thank you for pointing this out.

22.1  In equation (9), Δt represents the time step, which is used in the numerical integration method to help us calculate the change of the system state from t to Δt in the implicit Euler integration method.

22.2  The function f depends on t instead of t+Δt in the second line. This is because we use the implicit Euler integration method, which uses the information of the current time step t to make predictions and solve. The characteristic of this method is that it integrates the variables at the current moment, ensuring the numerical stability and unconditional convergence of the algorithm.

22.3   Regarding the difference in the time dependence of Fint and Fext: Fint is calculated based on the strain energy gradient of the system at time t+Δt, while Fext is the external force applied under known external conditions at time t. Since the internal force is related to the deformation state of the system, we solve it at time t+Δ. We have explained this point in detail in the text.

Comments 23: P. 11, l. 308. Superscript d appears without explanations.

Response 23: Thank you for pointing this out. We agree with this comment. Therefore, we have explained the meaning of the superscript d in the text. Here, Rd represents the d-dimensional Euclidean space where the object position vector x(t) is located, where d represents the dimension of the vector, that is, the position description of the system in two-dimensional space. These revisions can be found on page 11, from line 335-337.

Comments 24: P. 11, l. 314. It is unclear why the conventional numerical methods are not suitable for solving the equations of the breast model.

Response 24: Thank you for pointing this out. Traditional numerical methods such as Newton-Raphson and Runge-Kutta methods are often suitable for solving rigid systems. However, breast tissue has significant nonlinear and flexible characteristics, and the deformation behavior of breast tissue cannot be effectively simulated by these methods. Especially when dealing with large deformations and complex stress-strain relationships, the rigid system method will lead to numerical instability or insufficient solution accuracy.

Comments 25:  P. 11, l. 324. The authors introduce abbreviation ADMM but do not spell it out explicitly.

Response 25: Thank you for pointing this out. We agree with this comment. Therefore, we have added the full name of ADMM. These revisions can be found on page 12, from line 352.

Comments 26:  P. 12, l. 342. Parameter J appears unexplained.

Response 26: Thank you for pointing this out. We agree with this comment. Therefore, we have supplemented the meaning of parameter J, which is used to describe the change in volume during material deformation. These revisions can be found on page 13, from line 373-374.

Comments 27:  P. 12, l. 345. It is unclear what the authors mean under parameter I4.

Response 27: Thank you for pointing this out. We have added an explanation of parameter I4 in the text. Parameter I4 stands for the fourth invariant and is specifically used to describe the orientation and strain behavior of fiber materials (such as ligaments and connective tissue) in breast tissue. By introducing I4, we are able to more accurately simulate the anisotropic deformation characteristics of breast tissue, thereby achieving a more realistic simulation of the deformation behavior of breast tissue. Detailed explanation and calculation can be found in references [21-22].(page 13, paragraph, and line 378-379)

Comments 28:  P. 12, l. 360. The authors write W is the weighting matrix, but they have already used this notation in Eq. (12).

Response 28: Thank you for pointing this out. We agree with this comment. W as an expression appears twice in the article. Let's change one of the W's to G's. These revisions can be found on page 12, from Eq. (12).

Comments 29:  P. 13, l. 363. It is unclear which matrix in Eq. (19) the authors refer to.

Response 29: Thank you for pointing this out. In fact, Eq. (11) is solved using the method of Eq. (18). If you want to know more about the solution details, please refer to reference [19].

Comments 30:  P. 15, l. 412–417. The sentence given in these lines looks too challenging to read, and I recommend the authors rewrite it.

Response 30: Thank you for pointing this out. We agree with this comment. Therefore, we have rewritten it to be more concise and easier to understand.These revisions can be found on page 16, and line 453-459.

Comments 31: P. 15, l. 418. It is unclear what actuator the authors consider here.

Response 31: Thank you for pointing this out. We agree with this comment. The type of actuator mentioned is a stepper motor, which is used to drive the manipulation and adjustment of breast tissue. The actuator achieves precise manipulation of breast tissue through feedback control, ensuring that the system achieves the required displacement and accuracy during operation.

Comments 32: P. 15, Fig. 8. The figure contains unexplained notations yp and yd, and notation ypappears above two signal lines, which looks incorrect.

Response 32: Thank you for pointing this out. We agree with this comment. The symbols yp â€‹, yd, and yp`​in the figure represent the predicted output signal, control signal, and real position signal fed back by imaging. The position of yp` represents the real position signal measured by imaging technology, which is used to update the Jacobian matrix to ensure the accuracy of the control system.

3. Response to Comments on the Quality of English Language

Point 1: There are many typos or strange and unclear terms and phrases that should be revised (see comment no. 7).

Response 1:   Thank you for pointing this out. We agree with this comment. We have made modifications to question 7.

Reviewer 2 Report

Comments and Suggestions for Authors

1) Comparisons with other algorithms should be highlighted.

2)  The work is very interesting, and some related work should be discussed in the manuscript as the following. “(1) Cao, Y.; Li J.; Dong Z.; Sheng T.; Zhang D.; Cai J.; Jiang Y. Flexible tactile sensor with an embedded-hair-in-elastomer structure for normal and shear stress sensing. Soft. Sci. 2023, 3, 32. http://dx.doi.org/10.20517/ss.2023.22

3) The English representation requires more attention. The authors should check the English presentation throughout the manuscript.I believe more clear writing could help Authors to express their findings more effectively.

4) The parameters in Eqs. should be highlighted.

5) Figure 9 (Simulation result)  could be compared to experiment results or not?

6) Caption for Figure 10 is wrong, and it should be corrected.

Author Response

Comments 1: Comparisons with other algorithms should be highlighted.

Response 1: Thank you for pointing this out. We agree with this comment. We have clarified in pages 12, lines 348-354 of the article that the traditional method is slower, which is a critical limitation in surgical procedures, as medical surgeries require real-time precision. Traditional numerical methods (such as Newton-Raphson and Runge-Kutta methods) are generally more suitable for solving rigid systems. However, breast tissue exhibits significant nonlinear and flexible properties, making it difficult for these methods to accurately simulate its deformation. This challenge is especially pronounced when dealing with large deformations and complex stress-strain relationships, where applying rigid system methods can lead to numerical instability or insufficient solution accuracy.

Comments 2: The work is very interesting, and some related work should be discussed in the manuscript as the following. “(1) Cao, Y.; Li J.; Dong Z.; Sheng T.; Zhang D.; Cai J.; Jiang Y. Flexible tactile sensor with an embedded-hair-in-elastomer structure for normal and shear stress sensing. Soft. Sci. 2023, 3, 32. http://dx.doi.org/10.20517/ss.2023.22

Response 2: Thank you for pointing this out. We agree with this comment. Therefore, this paper is indeed interesting and relevant to this paper. We have cited the related work of Cao et al. (2023) in the paper and discussed it in the “Related Work” section. In the revised manuscript this change can be found – page 2, paragraph, and line 76-78.

Comments 3: The English representation requires more attention. The authors should check the English presentation throughout the manuscript.I believe more clear writing could help Authors to express their findings more effectively.

Response 3: Thank you for pointing this out. We agree with this comment. Therefore, we have carefully checked word and revised the grammar to improve the clarity and flow of the article.

Comments 4: The parameters in Eqs. should be highlighted.

Response 4: Thank you for pointing this out. We agree with this comment. Therefore, we have added parameters in appropriate places. For detailed parameters, please refer to the references in the corresponding chapters.

Comments 5:Figure 9 (Simulation result)  could be compared to experiment results or not?

Response 5: Thank you for pointing this out. Comparison between simulation results and experimental results can verify the reliability of the theory, so we believe that this comparison is necessary.

Comments 6: Caption for Figure 10 is wrong, and it should be corrected.

Response 6: Thank you for pointing this out. We agree with this comment. Therefore, we have corrected the incorrect text in Figure 12. In the revised manuscript this change can be found – page 19, paragraph, and line 528.

Reviewer 3 Report

Comments and Suggestions for Authors

This manuscript, entitled with “Study on Bionic Design and Tissue Manipulation of Breast Interventional Robot”, introduces a bionic breast interventional robot. The author simulated the morphology of the anterior chelae and the pedipalp with the sting of scorpions during their hunting process and designed this bionic robot. This bionic breast interventional robot is divided into a puncture module and a tissue manipulation module. By adjusting the breast shape, the accuracy of breast surgery intervention is improved, and a new solution is provided for the diagnosis and treatment of breast cancer. Nonetheless, several points need to be revised before the publication of this manuscript.

1.       In L432, the author mentions using pork as the experimental object for the experiment, but the structure and main components of pork and breast differ significantly. The author should supplement the explanation for choosing pork as the experimental object and discuss how the differences between pork tissue and breast tissue may affect the experimental results. Meanwhile, it is recommended to provide additional data or theoretical support to demonstrate that robots can achieve the same accuracy in breast tissue.

2.       In L427, the author mentions that in order to observe more clearly, suction cups are not used to fix tissues, only pushing and pulling tissues. In practical applications, will suction cup fixation of tissues cause breast deformation that is different from pushing and pulling tissues, thereby reducing accuracy. The author should consider using transparent suction cups to observe internal changes and simulate practical application scenarios simultaneously. If the author believes that the deformation caused by suction cup fixation and push-pull tissue is the same, corresponding evidence and theoretical explanations should be provided.

3.       As a key image showcasing the commonalities and design ideas between the bionic robot and scorpion, Figure 3's clarity and detail presentation are crucial for readers to understand the author's creative design. Suggest the author to enlarge Figure 3 as a whole and further display the specific design details of the robot, such as the connection between modules and the shape of key components, based on the enlargement.

4.       L82~L85, the author should reorganize this sentence to ensure clear meaning.

5.       In order to comprehensively evaluate the performance and accuracy of robots in different situations, it is recommended that the author change the position of targets and obstacles in the prosthesis experiment and conduct multiple experiments. At the same time, experimental data should be detailed and analyzed to fully verify the performance of robots in various complex situations. This experimental design will be closer to practical application scenarios and provide more reliable data support for the clinical application of robots.

6.       Considering that the patient's breast may have pathological changes due to breast cancer or other diseases, it is recommended that the author further establish a breast model in the state of disease on the basis of the silicone model. By adding such experimental content, the performance of robots in dealing with different pathological conditions can be more comprehensively evaluated, thereby improving their adaptability and reliability in clinical applications.

7.       The purpose of surgery is to alleviate the patient's pain, which is a very serious matter. While this robot significantly improves the accuracy of breast intervention surgery, it cannot guarantee that puncture will not fail due to the individual patient variability. Therefore, it is recommended to design a mechanism that allows doctors to confirm before surgery and adjust or stop the procedure at any time during the puncture process. This will help ensure the safety of the surgery and fully respect the rights of patients and medical ethical principles.

8.       The sterile environment during the surgical process is a key factor in ensuring surgical safety. The author should give sufficient consideration and explanation to the selection of robot materials and their preoperative disinfection issues.

Author Response

Comments 1: In L432, the author mentions using pork as the experimental object for the experiment, but the structure and main components of pork and breast differ significantly. The author should supplement the explanation for choosing pork as the experimental object and discuss how the differences between pork tissue and breast tissue may affect the experimental results. Meanwhile, it is recommended to provide additional data or theoretical support to demonstrate that robots can achieve the same accuracy in breast tissue.

Response 1: Thank you for pointing this out. We agree with this comment. Thank you for your valuable comments on the choice of our experimental model. We chose pork as the experimental subject mainly because it has similar properties to human tissue in terms of elasticity and force deformation. Although we acknowledge that there are differences in structure and composition between pork and breast tissue, the focus of this experiment is to evaluate the mechanical control and precision of the robotic system, and these characteristics can be better simulated and verified by pork tissue. We also agree that the differences between pork and breast tissue may affect the tissue interactions in the experimental results. In order to further verify the performance of the system in breast tissue, we plan to use human tissue analog materials or models that are closer to biological conditions in future work. We supplement it at line 465-469. 

Comments 2: In L427, the author mentions that in order to observe more clearly, suction cups are not used to fix tissues, only pushing and pulling tissues. In practical applications, will suction cup fixation of tissues cause breast deformation that is different from pushing and pulling tissues, thereby reducing accuracy. The author should consider using transparent suction cups to observe internal changes and simulate practical application scenarios simultaneously. If the author believes that the deformation caused by suction cup fixation and push-pull tissue is the same, corresponding evidence and theoretical explanations should be provided.

Response 2: Thank you for pointing this out. Suction cup fixation of breast tissue does cause some deformation, but we believe that the impact of this deformation in actual operation can be corrected through a real-time feedback mechanism. Although the deformation caused by suction cup fixation and push-pull operation is different in degree, the impact on target positioning and path planning is basically the same. We are studying how to further quantify the difference between the two methods, and plan to introduce transparent suction cups in future work to better simulate actual operation scenarios and provide more direct data support.

Comments 3: As a key image showcasing the commonalities and design ideas between the bionic robot and scorpion, Figure 3's clarity and detail presentation are crucial for readers to understand the author's creative design. Suggest the author to enlarge Figure 3 as a whole and further display the specific design details of the robot, such as the connection between modules and the shape of key components, based on the enlargement.

Response 3: Thank you for pointing this out. We acknowledge that your suggestion is good. However, Figure 3 of this article is only a conceptual diagram, which will be studied in our future work. The purpose of this article is to verify target manipulation, so we did not modify Figure 3 this time. If you think that our thought is not good, please contact us in time, and we will make the modification as soon as possible and listen to your suggestions.

Comments 4: L82~L85, the author should reorganize this sentence to ensure clear meaning.

Response 4: Thank you for pointing this out. We agree with this comment. Therefore, we have reworded the statement to make it clear and concise.The revised manuscript this change can be found – page 2, paragraph, and line 85-88.

Comments 5: In order to comprehensively evaluate the performance and accuracy of robots in different situations, it is recommended that the author change the position of targets and obstacles in the prosthesis experiment and conduct multiple experiments. At the same time, experimental data should be detailed and analyzed to fully verify the performance of robots in various complex situations. This experimental design will be closer to practical application scenarios and provide more reliable data support for the clinical application of robots.

Response 5: Thank you for pointing this out. We agree with this comment. Therefore, we added 2 targets on the basis of the original experiment. This revision can be found in Figure 14.

Comments 6:  Considering that the patient's breast may have pathological changes due to breast cancer or other diseases, it is recommended that the author further establish a breast model in the state of disease on the basis of the silicone model. By adding such experimental content, the performance of robots in dealing with different pathological conditions can be more comprehensively evaluated, thereby improving their adaptability and reliability in clinical applications.

Response 6: Thank you for pointing this out. We agree with this comment. Therefore, we have fully understand the importance of establishing breast models under pathological conditions and agree that this will help further verify the clinical suitability of the robot. However, based on the current experimental conditions and data collection limitations, it is temporarily impossible to establish and verify these models. Nevertheless, our experiments have fully demonstrated the performance of the robot under different conditions and laid the foundation for future clinical applications. In future studies, we plan to introduce more models and data related to pathological conditions to further evaluate the performance and reliability of the robot. Thank you again for your careful feedback and valuable suggestions.

Comments 7:  The purpose of surgery is to alleviate the patient's pain, which is a very serious matter. While this robot significantly improves the accuracy of breast intervention surgery, it cannot guarantee that puncture will not fail due to the individual patient variability. Therefore, it is recommended to design a mechanism that allows doctors to confirm before surgery and adjust or stop the procedure at any time during the puncture process. This will help ensure the safety of the surgery and fully respect the rights of patients and medical ethical principles.

Response 7: Thank you for pointing this out. We agree with this comment. Your suggestion is correct and necessary. However, our experiment is still in the laboratory stage, and we will continue to improve it so that it can better serve patients. The current robot can be stopped at any time to respect the patient's wishes. In the future, we will definitely add the mechanism you suggested.

Comments 8:  The sterile environment during the surgical process is a key factor in ensuring surgical safety. The author should give sufficient consideration and explanation to the selection of robot materials and their preoperative disinfection issues.

Response 8: Thank you for pointing this out. We agree with this comment. Since this experiment is in the laboratory stage, we are currently focusing more on the accuracy of the experimental results. We will address the final material selection and preoperative disinfection of the robot in future work.

Reviewer 4 Report

Comments and Suggestions for Authors

In this work, the authors proposed a bionic breast interventional robot that mimics the scorpion's predation process. The method of active manipulation of tissue deformation was simulated using MATALB software, and the experiments were conducted. This is an interesting paper on developing bionic robot. However, I cannot recommend the publication unless the authors can address the following concerns.

 1. There is a mistake about the directions of the rectangular coordinate system in Figure 3.

 2. The difference of the designed bionic robot and the manufactured prototype should be provided.

 3. Since the patient lies prone on the bed in the design, why the breast prosthesis and in vitro tissue were not placed prone in the simulation and experiments of section 4?

 5. How the target point was observed in vitro tissue, as it is not transparent?

Comments on the Quality of English Language

There is no comment about the English Language.

Author Response

Comments 1:There is a mistake about the directions of the rectangular coordinate system in Figure 3.

Response 1: Thank you for pointing this out. We agree with this comment. Thank you for pointing out the problem with the orientation of the coordinate system of the rectangle in Figure 3. We have revised the figure to ensure that the orientation of the coordinate system is correct. Thank you again for your thoughtful feedback.

Comments 2: The difference of the designed bionic robot and the manufactured prototype should be provided.

Response 2: Thank you for pointing this out. We agree with this comment. We have explained in the paper that only a single breast tissue was studied in the experiment to verify the theory, so the actual tissue manipulation module is different from the tissue manipulation module of the virtual prototype. Specifically, the tissue manipulation module has three degrees of freedom (DOF) for manipulating a single breast tissue instead of six DOF. To avoid misunderstanding, we have re-expressed line 128-147 and 170-188.

Comments 3: Since the patient lies prone on the bed in the design, why the breast prosthesis and in vitro tissue were not placed prone in the simulation and experiments of section 4?

Response 3: Thank you for pointing this out. We agree with this comment. We did not place the breast prosthesis and in vitro tissue in a prone position in the experiment, mainly to better observe and record the displacement changes of the internal target points. This experimental design can more clearly demonstrate the effects of tissue manipulation and the precise displacement of the target points, thereby ensuring the visualization and analysis of the experimental data. In future work, we plan to conduct more experiments to simulate situations that are closer to actual surgical scenarios, including considering prone placement to evaluate the impact of this method in practical applications. We have also expressed this in 436-437 of the original text. If you are dissatisfied with our response, please contact us in time and we will make the revisions as soon as possible.

Comments 5: How the target point was observed in vitro tissue, as it is not transparent?

Response 5: Thank you for pointing this out. We agree with this comment. Therefore, in our experiment, we used a CCD camera and image processing system to infer the displacement changes of target points in opaque in vitro tissues through surface markers. Although there are errors in this process, it is mainly to verify the effect inside the ex vivo tissue. In future work, we plan to conduct more experiments to overcome this difficulty.

Round 2

Reviewer 1 Report

Comments and Suggestions for Authors

The authors have revised the paper and improved its quality, but some minor technical issues have remained:

1.        P. 10, l. 283. Matrix notation 63T still looks inappropriate here. The authors equate this matrix to the 4 × 1 column vector, which looks inaccurate from the math point of view.

2.        P. 10, Eq. (5). This equation still looks inaccurate. On the right side of the equation, the authors multiple two 4 × 1 column vectors, which is an infeasible operation. Isn’t the last multiplier (which repeats the left side of the equation) redundant here?

3.        P. 10, Eq. (6). The dots in the last column over the p variables still look inappropriate. Shouldn’t there be the prime signs?

4.        P. 19–20, Eqs. (35) and (37). What do superscripts f in the right side of each equation mean here?

Author Response

Comments 1: P. 10, l. 283. Matrix notation 63T still looks inappropriate here. The authors equate this matrix to the 4 × 1 column vector, which looks inaccurate from the math point of view.

Response 1: Thank you for pointing this out. We agree with this comment. Matrix 63T is indeed incorrect here, and we have revised it. Setting the 63T letter here is not appropriate by itself, so I changed 63T to 63A and added the statement: Set the position of the translation probe into joint 6 relative to joint 3 to 63A so that 63A represents the column vector with relative position 4*1, avoiding the ambiguity expressed by 63T. In the revised manuscript this change can be found – page 10, paragraph, and line 281-282, line 283-285.

Comments 2: P. 10, Eq. (5). This equation still looks inaccurate. On the right side of the equation, the authors multiple two 4 × 1 column vectors, which is an infeasible operation. Isn’t the last multiplier (which repeats the left side of the equation) redundant here?

Response 2: Thank you for pointing this out. We agree with this comment. Therefore, equation (5) does contain mathematical errors. Equation (4) is modified according to the modification in reply to reviewer's comment 1, so that the expression of equation (4) should be the conversion relationship between (px,py,pz,1) and (px',py',pz',1). The improvement of equation (5) is further derived based on equation (4) and 60A=30A*63A, which we have also modified. In the revised manuscript this change can be found – page 10, paragraph, and Equation (4),Equation (5).

Comments 3: P. 10, Eq. (6). The dots in the last column over the p variables still look inappropriate. Shouldn’t there be the prime signs?

Response 3: Thank you for pointing this out. We agree with this comment. The labeling of equation (6) p. is inconsistent with p' of other equations, which we have modified.In the revised manuscript this change can be found – page 10, paragraph, and Equation (6).

Comments 4: P. 19–20, Eqs. (35) and (37). What do superscripts f in the right side of each equation mean here?

Response 4: Thank you for pointing this out. We agree with this comment. xcf, ycf and zcf are the homogeneous coordinates of the space point in the camera coordinate, and also the homogeneous coordinates of the image point in the image coordinate system. In the revised manuscript this change can be found – page 19, paragraph, and line 536-538.

Reviewer 2 Report

Comments and Suggestions for Authors

All the comments are well revised, and it could be accepted for publication.

Author Response

Comments 1: All the comments are well revised, and it could be accepted for publication.

Response 1: Thank you for pointing this out. We agree with this comment. Thank you for your recognition and thank you again for your valuable comments.

Reviewer 4 Report

Comments and Suggestions for Authors

The authors have carefully revised the manuscript according to the reviewer's comments. There are no more new questions or comments for the authors. Therefore, I agree to recommend this paper for publication.

Author Response

Comments 1: The authors have carefully revised the manuscript according to the reviewer's comments. There are no more new questions or comments for the authors. Therefore, I agree to recommend this paper for publication.

Response 1: Thank you for pointing this out. We agree with this comment. Thank you for your recognition and thank you again for your valuable comments.
